# Speeding Up Latent Variable Gaussian Graphical Model Estimation via Nonconvex Optimization

**Pan Xu**
Department of Computer Science
University of Virginia
Charlottesville, VA 22904
px3ds@virginia.edu

**Jian Ma**
School of Computer Science
Carnegie Mellon University
Pittsburgh, PA 15213
jianma@cs.cmu.edu

**Quanquan Gu**
Department of Computer Science
University of Virginia
Charlottesville, VA 22904
qg5w@virginia.edu

## Abstract

We study the estimation of the latent variable Gaussian graphical model (LVGGM), where the precision matrix is the superposition of a sparse matrix and a low-rank matrix. In order to speed up the estimation of the sparse plus low-rank components, we propose a sparsity constrained maximum likelihood estimator based on matrix factorization, and an efficient alternating gradient descent algorithm with hard thresholding to solve it. Our algorithm is orders of magnitude faster than the convex relaxation based methods for LVGGM. In addition, we prove that our algorithm is guaranteed to linearly converge to the unknown sparse and low-rank components up to the optimal statistical precision. Experiments on both synthetic and genomic data demonstrate the superiority of our algorithm over the state-of-the-art algorithms and corroborate our theory.

## 1 Introduction

For a $d$-dimensional Gaussian graphical model (i.e., multivariate Gaussian distribution) $N(\mathbf{0}, \boldsymbol{\Sigma}^*)$, the inverse of covariance matrix $\boldsymbol{\Omega}^* = (\boldsymbol{\Sigma}^*)^{-1}$ (also known as the precision matrix or concentration matrix) measures the conditional dependence relationship between marginal random variables [19]. When the number of observations is comparable to the ambient dimension of the Gaussian graphical model, additional structural assumptions are needed for consistent estimation. Sparsity is one of the most common structures imposed on the precision matrix in Gaussian graphical models (GGM), because it gives rise to a sparse graph, which characterizes the conditional dependence of the marginal variables. The problem of estimating the sparse precision matrix in Gaussian graphical models has been studied by a large body of literature [23, 29, 12, 28, 6, 34, 37, 38, 33]. However, the real world data may not follow a sparse GGM, especially when some of the variables are unobservable.

To alleviate this problem, the latent variable Gaussian graphical model (LVGGM) [9, 24] has been studied, where the precision matrix of the observed variables is conditionally sparse given the latent variables (i.e., unobserved), but marginally not sparse. It is well-known that in LVGGM, the precision matrix $\boldsymbol{\Omega}^*$ can be represented as the superposition of a sparse matrix $\mathbf{S}^*$ and a low-rank matrix $\mathbf{L}^*$, where the latent variables contribute to the low rank component in the precision matrix. In other words, we have $\boldsymbol{\Omega}^* = \mathbf{S}^* + \mathbf{L}^*$.

In the learning problem of LVGGM, the goal is to estimate both the unknown sparse component $\mathbf{S}^*$ and the low-rank component $\mathbf{L}^*$ of the precision matrix simultaneously. In the seminal work,

Chandrasekaran et al. [9] proposed a maximum-likelihood estimator based on $\ell_1$ norm penalty on the sparse matrix and nuclear norm penalty on the low-rank matrix, and proved the model selection consistency for LVGGM estimation. Meng et al. [24] studied a similar penalized estimator, and derived Frobenius norm error bounds based on the restricted strong convexity [26] and the structural Fisher incoherence condition between the sparse and low-rank components. Both of these two methods for LVGGM estimation are based on a penalized convex optimization problem, which can be solved by log-determinant proximal point algorithm [32] and alternating direction method of multipliers [22]. Due to the nuclear norm penalty, these convex optimization algorithms need to do full singular value decomposition (SVD) to solve the proximal mapping of nuclear norm at each iteration, which results in an extremely high time complexity of $O(d^3)$. When $d$ is large as often in the high dimensional setting, the convex relaxation based methods are computationally intractable. It is worth noting that full SVD cannot be accelerated by power method [13] or other randomized SVD algorithms [15], hence the $O(d^3)$ is unavoidable whenever nuclear norm regularization is employed.

In this paper, in order to speed up learning LVGGM, we propose a novel sparsity constrained maximum likelihood estimator for LVGGM based on matrix factorization. Specifically, inspired by the recent work on matrix factorization [18, 16, 44, 45, 11, 30], we propose to reparameterize the low-rank component $\mathbf{L}$ in the precision matrix as the product of smaller matrices, i.e., $\mathbf{L} = \mathbf{Z}\mathbf{Z}^\top$, where $\mathbf{Z} \in \mathbb{R}^{d \times r}$ and $r \leq d$ is the number of latent variables. This factorization captures the intrinsic low-rank structure of $\mathbf{L}$, and automatically ensures its low-rankness. We propose an alternating gradient descent with hard thresholding to solve the new estimator. We prove that the output of our algorithm is guaranteed to linearly converge to the unknown parameters up to the statistical precision. In detail, our algorithm enjoys $O(d^2 r)$ per-iteration time complexity, which outperforms the $O(d^3)$ per-iteration complexity of state-of-the-art LVGGM estimators based on nuclear norm penalty [9, 22, 24]. In addition, the estimators from our algorithm for LVGGM attain $\max\{O_p(\sqrt{s^* \log d / n}), O_p(\sqrt{rd/n})\}$ statistical rate of convergence in terms of Frobenius norm, where $s^*$ is the conditional sparsity of the precision matrix (i.e., sparsity of $\mathbf{S}^*$), and $r$ is the number of latent variables (i.e., rank of $\mathbf{L}^*$). This matches the minimax optimal convergence rate for LVGGM estimation [9, 1, 24]. Thorough experiments on both synthetic and breast cancer genomic datasets show that our algorithm is orders of magnitude faster than existing methods.

It is also worth noting that, although our estimator and algorithm is designed for LVGGM, it is directly applicable to the Gaussian graphical model where the precision matrix is the sum of a sparse matrix and a low-rank matrix. And the theoretical guarantees of our algorithm still hold.

The remainder of this paper is organized as follows: In Section 2, we briefly review existing work that is relevant to our study. We present our estimator and algorithm in detail in Section 3, and the main theory in Section 4. In Section 5, we compare the proposed algorithm with the state-of-the-art algorithms on both synthetic data and real-world breast cancer data. Finally, we conclude this paper in Section 6.

**Notation** For matrices $\mathbf{A}, \mathbf{B}$ with commensurate dimensions, we use $\langle \mathbf{A}, \mathbf{B} \rangle = \mathrm{tr}(\mathbf{A}^\top \mathbf{B})$ to denote their inner product and $\mathbf{A} \otimes \mathbf{B}$ denote their Kronecker product. For a matrix $\mathbf{A} \in \mathbb{R}^{d \times d}$, we denote its (ordered) singular values by $\sigma_1(\mathbf{A}) \geq \sigma_2(\mathbf{A}) \geq \ldots \geq \sigma_d(\mathbf{A}) \geq 0$. We denote by $\mathbf{A}^{-1}$ the inverse of $\mathbf{A}$, and denote by $|\mathbf{A}|$ its determinant. We use the notation $\| \cdot \|$ for various types of matrix norms, including the spectral norm $\|\mathbf{A}\|_2$ and the Frobenius norm $\|\mathbf{A}\|_F$. We also use the following norms $\|\mathbf{A}\|_{0,0} = \sum_{i,j} \mathbb{1}(A_{ij} \neq 0)$, $\|\mathbf{A}\|_{\infty,\infty} = \max_{1 \leq i,j \leq d} |A_{ij}|$, and $\|\mathbf{A}\|_{1,1} = \sum_{i,j=1}^{d} |A_{ij}|$. A constant is called absolute constant if it does not depend on the parameters of the problem, e.g., dimension and sample size. We denote $a \lesssim b$ if $a$ is less than $b$ up to a constant.

## 2 Additional Related Work

Precision matrix estimation in sparse Gaussian graphical models (GGM) is commonly formulated as a penalized maximum likelihood estimation problem with $\ell_{1,1}$ norm regularization [12, 29, 28] (graphical Lasso) or regularization on diagonal elements of Cholesky decomposition for precision matrix [17]. Due to the complex dependency among marginal variables in many applications, sparsity assumption on the precision matrix often does not hold. To relax this assumption, the conditional Gaussian graphical model (cGGM) was proposed in [41, 5] and the partial Gaussian graphical model (pGGM) was proposed in [42], both of which impose blockwise sparsity on the precision matrix and estimate multiple blocks therein. Despite a good interpretation of these models, they need to access both the observed variables as well as the latent variables for estimation. Another alternative

is the latent variable Gaussian graphical model (LVGGM), which was proposed in [9], and later investigated in [22, 24]. Compared with cGGM and pGGM, the estimation of LVGGM does not need to access the latent variables and therefore is more flexible.

Another line of research related to ours is low-rank matrix estimation based on alternating minimization and gradient descent [18, 16, 44, 45, 11, 30, 3, 35, 43]. However, extending them to low-rank and sparse matrix estimation as in LVGGM turns out to be highly nontrivial. The most related work to ours includes [14] and [40], which studied nonconvex optimization for low-rank plus sparse matrix estimation. However, they are limited to robust PCA [8] and multi-task regression [1] in the noiseless setting. Due to the square loss in RPCA, the sparse matrix $\mathbf{S}$ can be calculated by subtracting the low-rank matrix $\mathbf{L}$ from the observed data matrix. Nevertheless, in LVGGM, there is no closed-form solution for the sparse matrix due to the log-determinant term, and we need to use gradient descent to update $\mathbf{S}$. On the other hand, both the algorithm in [40] and our algorithm have an initialization stage. Yet our initialization algorithm is new and different from the initialization algorithm in [40] for RPCA. Furthermore, our analysis of the initialization algorithm is built on the spikiness condition, which is also different from that for RPCA.

The last but not least related work is expectation maximization (EM) algorithm [2, 36], which shares a similar bivariate structure as our estimator. However, the proof technique used in [2, 36] is not directly applicable to our algorithm, due to the matrix factorization structure in our estimator. Moreover, to overcome the dependency issue between consecutive iterations in the proof, sample splitting strategy [18, 16] was employed in [2, 36, 39], i.e., dividing the whole dataset into $T$ pieces and using a fresh piece of data in each iteration. Unfortunately, the sample splitting technique results in a suboptimal statistical rate, incurring an extra factor of $\sqrt{T}$ in the rate. In sharp contrast, our proof technique does not rely on sample splitting, because we are able to prove a uniform convergence result over a small neighborhood of the unknown parameters, which directly resolves the dependency issue.

## 3 The Proposed Estimator and Algorithm

In this section, we present a new estimator for LVGGM estimation, together with a new algorithm.

### 3.1 Latent Variable GGMs

Let $\boldsymbol{X}_O$ be the $d$-dimensional random vector with observed variables and $\boldsymbol{X}_L$ be the $r$-dimensional random vector with latent variables. We assume that the concatenated random vector $\boldsymbol{X} = (\boldsymbol{X}_O^\top, \boldsymbol{X}_L^\top)^\top$ follows a multivariate Gaussian distribution with covariance matrix $\widetilde{\boldsymbol{\Sigma}}$ and sparse precision matrix $\widetilde{\boldsymbol{\Omega}} = \widetilde{\boldsymbol{\Sigma}}^{-1}$. It is proved in [10] that the observed data $\boldsymbol{X}_O$ follows a normal distribution with marginal covariance matrix $\boldsymbol{\Sigma}^* = \widetilde{\boldsymbol{\Sigma}}_{OO}$, which is the top-left block matrix in $\widetilde{\boldsymbol{\Sigma}}$ corresponding to $\boldsymbol{X}_O$. The precision matrix of $\boldsymbol{X}_O$ is then given by Schur complement [13]:

$$\boldsymbol{\Omega}^* = (\widetilde{\boldsymbol{\Sigma}}_{OO})^{-1} = \widetilde{\boldsymbol{\Omega}}_{OO} - \widetilde{\boldsymbol{\Omega}}_{OL}\widetilde{\boldsymbol{\Omega}}_{LL}^{-1}\widetilde{\boldsymbol{\Omega}}_{LO}. \tag{3.1}$$

Since we can only observe $\boldsymbol{X}_O$, the marginal precision matrix $\boldsymbol{\Omega}^*$ is generally not sparse. We define $\mathbf{S}^* := \widetilde{\boldsymbol{\Omega}}_{OO}$ and $\mathbf{L}^* := -\widetilde{\boldsymbol{\Omega}}_{OL}\widetilde{\boldsymbol{\Omega}}_{LL}^{-1}\widetilde{\boldsymbol{\Omega}}_{LO}$. Then $\mathbf{S}^*$ is sparse due to the sparsity of $\widetilde{\boldsymbol{\Omega}}$. We do not impose any dependency restriction on $\boldsymbol{X}_O$ and $\boldsymbol{X}_L$, and thus the matrices $\widetilde{\boldsymbol{\Omega}}_{OL}$ and $\widetilde{\boldsymbol{\Omega}}_{LO}$ can be potentially dense. We assume that the number of latent variables is smaller than that of the observed. Therefore, $\mathbf{L}^*$ is low-rank and may be dense. Thus, the precision matrix of LVGGM can be written as

$$\boldsymbol{\Omega}^* = \mathbf{S}^* + \mathbf{L}^*, \tag{3.2}$$

where $\|\mathbf{S}^*\|_{0,0} = s^*$ and $\mathrm{rank}(\mathbf{L}^*) = r$. We refer to [9] for a detailed discussion of LVGGM.

### 3.2 The Proposed Estimator

Suppose that we observe i.i.d. samples $\boldsymbol{X}_1, \ldots, \boldsymbol{X}_n$ from $N(\mathbf{0}, \boldsymbol{\Sigma}^*)$. Our goal is to estimate the sparse component $\mathbf{S}^*$ and the low-rank component $\mathbf{L}^*$ of the unknown precision matrix $\boldsymbol{\Omega}^*$ in (3.2). The negative log-likelihood of the Gaussian graphical model is proportional to the following sample loss function up to a constant

$$p_n(\mathbf{S}, \mathbf{L}) = \mathrm{tr}\left[\widehat{\boldsymbol{\Sigma}}\big(\mathbf{S} + \mathbf{L}\big)\right] - \log|\mathbf{S} + \mathbf{L}|, \tag{3.3}$$

where $\widehat{\boldsymbol{\Sigma}} = 1/n\sum_{i=1}^n \boldsymbol{X}_i\boldsymbol{X}_i^\top$ is the sample covariance matrix, and $|\mathbf{S} + \mathbf{L}|$ is the determinant of $\boldsymbol{\Omega} = \mathbf{S} + \mathbf{L}$. We employ the maximum likelihood principle to estimate $\mathbf{S}^*$ and $\mathbf{L}^*$, which is equivalent to minimizing the negative log-likelihood in (3.3).

The low-rank structure of the precision matrix, i.e., $\mathbf{L}$ poses a great challenge for computation. A typical way is to use nuclear-norm regularized estimator, or rank constrained estimator to estimate $\mathbf{L}$. However, such kind of estimators involve singular value decomposition at each iteration, which is computationally very expensive. To overcome this computational obstacle, we reparameterize $\mathbf{L}$ as the product of smaller matrices. More specifically, due to the symmetry of $\mathbf{L}$, it can be reparameterized by $\mathbf{L} = \mathbf{Z}\mathbf{Z}^\top$, where $\mathbf{Z} \in \mathbb{R}^{d \times r}$ and $r > 0$ is the number of latent variables and is a tuning parameter. This kind of reparametrization has recently been used in low-rank matrix estimation [18, 16, 44, 45, 11, 30] based on matrix factorization. Then we can rewrite the sample loss function in (3.3) as the following objective function

$$q_n(\mathbf{S}, \mathbf{Z}) = \mathrm{tr}\left[\widehat{\boldsymbol{\Sigma}}\big(\mathbf{S} + \mathbf{Z}\mathbf{Z}^\top\big)\right] - \log|\mathbf{S} + \mathbf{Z}\mathbf{Z}^\top|. \tag{3.4}$$

Based on (3.4), we propose a nonconvex estimator using sparsity constrained maximum likelihood:

$$\min_{\mathbf{S},\mathbf{Z}} \quad q_n(\mathbf{S}, \mathbf{Z}) \quad \text{subject to } \|\mathbf{S}\|_{0,0} \le s, \tag{3.5}$$

where $s > 0$ is a tuning parameter that controls the sparsity of $\mathbf{S}$.

### 3.3 The Proposed Algorithm

Due to the matrix factorization based reparameterization $\mathbf{L} = \mathbf{Z}\mathbf{Z}^\top$, the objective function in (3.5) is nonconvex. In addition, the sparsity based constraint in (3.5) is nonconvex as well. Therefore, the estimation in (3.5) is essentially a nonconvex optimization problem. We propose to solve it by alternately performing gradient descent with respect to one parameter matrix with the other one fixed. The detailed algorithm is displayed in Algorithm 1, which consists of two stages.

In the initialization stage (**Stage I**), it outputs initial points $\widehat{\mathbf{S}}^{(0)}, \widehat{\mathbf{Z}}^{(0)}$, which, we will show later, are guaranteed to fall in the small neighborhood of $\mathbf{S}^*$ and $\mathbf{Z}^*$ respectively. Note that we need to do inversion in Line 3, whose complexity is $O(d^3)$. Nevertheless, we only need to do inversion once. In sharp contrast, convex relaxation approaches need to do full SVD with $O(d^3)$ complexity at each iteration, which is much more time consuming than ours.

In the alternating gradient descent stage (**Stage II**), we iteratively estimate $\mathbf{S}$ while fixing $\mathbf{Z}$, and then estimate $\mathbf{Z}$ while fixing $\mathbf{S}$. Instead of solving each subproblem exactly, we propose to perform one-step gradient descent for $\mathbf{S}$ and $\mathbf{Z}$ alternately, using step sizes $\eta$ and $\eta'$. In Lines 6 and 8 of Algorithm 1, $\nabla_{\mathbf{S}} q_n(\mathbf{S}, \mathbf{Z})$ and $\nabla_{\mathbf{Z}} q_n(\mathbf{S}, \mathbf{Z})$ denote the partial gradient of $q_n(\mathbf{S}, \mathbf{Z})$ with respect to $\mathbf{S}$ and $\mathbf{Z}$ respectively. The choice of the step sizes will be clear according to our theory. In practice, one can also use line search to choose the step sizes. Due to the sparsity constraint $\|\mathbf{S}\|_{0,0} \le s$, we apply hard thresholding [4] right after the gradient descent step for $\mathbf{S}$, in Line 7 of Algorithm 1. For a matrix $\mathbf{S} \in \mathbb{R}^{d \times d}$ and an integer $s$, the hard thresholding operator $\mathcal{HT}_s(\mathbf{S})$ preserves the $s$ largest magnitudes in $\mathbf{S}$ and sets the rest entries to zero. Algorithm 1 does not involve singular value decomposition in each iteration, neither solve an exact optimization problem, which makes it much faster than the convex relaxation based algorithms [9, 24]. The computational overhead of Algorithm 1 mainly comes from the calculation of the partial gradient with respect to $\mathbf{Z}$, whose time complexity is $O(rd^2)$. Therefore, our algorithm has a per-iteration complexity of $O(rd^2)$.

## 4 Main Theory

We present our main theory in this section, which characterizes the convergence rate of Algorithm 1, and the statistical rate of its output. We begin with some definitions and assumptions, which are necessary for our theoretical analysis.

**Assumption 4.1.** There is a constant $\nu > 0$ such that $0 < 1/\nu \le \lambda_{\min}(\boldsymbol{\Sigma}^*) \le \lambda_{\max}(\boldsymbol{\Sigma}^*) \le \nu < \infty$, where $\lambda_{\min}(\boldsymbol{\Sigma}^*)$ and $\lambda_{\max}(\boldsymbol{\Sigma}^*)$ are the minimal and maximal eigenvalues of $\boldsymbol{\Sigma}^*$ respectively.

Assumption 4.1 requires the eigenvalues of true covariance matrix $\boldsymbol{\Sigma}^*$ to be finite and bounded below from a positive number, which is a standard assumption for Gaussian graphical models [29, 21, 28]. The relation between the covariance matrix and the precision matrix $\boldsymbol{\Omega}^* = (\boldsymbol{\Sigma}^*)^{-1}$ immediately yields $1/\nu \le \lambda_{\min}(\boldsymbol{\Omega}^*) \le \lambda_{\max}(\boldsymbol{\Omega}^*) \le \nu$.

It is well understood that the estimation problem of the decomposition $\boldsymbol{\Omega}^* = \mathbf{S}^* + \mathbf{L}^*$ can be ill-posed, where identifiability issue arises when the low-rank matrix $\mathbf{L}^*$ is also sparse [10, 7]. The concept of *incoherence* condition, which was originally proposed for matrix completion [7], has been adopted in [9, 10], which ensures the low-rank matrix not to be too sparse by restricting the degree of coherence

---
**Algorithm 1** Alternating Thresholded Gradient Descent (AltGD) for LVGGM
---
1: **Input:** i.i.d. samples $\boldsymbol{X}_1, \ldots, \boldsymbol{X}_n$ from LVGGM, max number of iterations $T$, and parameters $\eta, \eta', r, s$.
   **Stage I: Initialization**
2: $\widehat{\boldsymbol{\Sigma}} = \frac{1}{n} \sum_{i=1}^{n} \boldsymbol{X}_i \boldsymbol{X}_i^\top$.
3: $\widehat{\mathbf{S}}^{(0)} = \mathcal{HT}_s(\widehat{\boldsymbol{\Sigma}}^{-1})$, which preserves the $s$ largest magnitudes of $\widehat{\boldsymbol{\Sigma}}^{-1}$.
4: Compute SVD: $\widehat{\boldsymbol{\Sigma}}^{-1} - \widehat{\mathbf{S}}^{(0)} = \mathbf{U}\mathbf{D}\mathbf{U}^\top$, where $\mathbf{D}$ is a diagonal matrix. Let $\widehat{\mathbf{Z}}^{(0)} = \mathbf{U}\mathbf{D}_r^{1/2}$, where $\mathbf{D}_r$ is the first $r$ columns of $\mathbf{D}$.

---
   **Stage II: Alternating Gradient Descent**
5: **for** $t = 0, \ldots, T-1$ **do**
6: $\quad \widehat{\mathbf{S}}^{(t+0.5)} = \widehat{\mathbf{S}}^{(t)} - \eta \nabla_{\mathbf{S}} q_n(\widehat{\mathbf{S}}^{(t)}, \widehat{\mathbf{Z}}^{(t)})$;
7: $\quad \widehat{\mathbf{S}}^{(t+1)} = \mathcal{HT}_s(\widehat{\mathbf{S}}^{(t+0.5)})$, which preserves the $s$ largest magnitudes of $\widehat{\mathbf{S}}^{(t+0.5)}$;
8: $\quad \widehat{\mathbf{Z}}^{(t+1)} = \widehat{\mathbf{Z}}^{(t)} - \eta' \nabla_{\mathbf{Z}} q_n(\widehat{\mathbf{S}}^{(t)}, \widehat{\mathbf{Z}}^{(t)})$;
9: **end for**
10: **output:** $\widehat{\mathbf{S}}^{(T)}, \widehat{\mathbf{Z}}^{(T)}$.
---

between singular vectors and the standard basis. Later work such as [1, 25] relaxed this condition to a constraint on the *spikiness ratio*, and showed that *spikiness* condition is milder than incoherence condition. In our theory, we use the notion of *spikiness* as follows.

**Assumption 4.2** (Spikiness Condition [25])**.** For a matrix $\mathbf{L} \in \mathbb{R}^{d \times d}$, the *spikiness ratio* is defined as $\alpha_{sp}(\mathbf{L}) := d\|\mathbf{L}\|_{\infty,\infty}/\|\mathbf{L}\|_F$. For the low-rank matrix $\mathbf{L}^*$ in (3.2), we assume that there exists a constant $\alpha^* > 0$ such that

$$\|\mathbf{L}^*\|_{\infty,\infty} = \frac{\alpha_{sp}(\mathbf{L}^*) \cdot \|\mathbf{L}^*\|_F}{d} \le \frac{\alpha^*}{d}. \tag{4.1}$$

Since $\text{rank}(\mathbf{L}^*) = r$, we define $\sigma_{\max} = \sigma_1(\mathbf{L}^*) > 0$ and $\sigma_{\min} = \sigma_r(\mathbf{L}^*) > 0$ to be the maximal and minimal nonzero singular value of $\mathbf{L}^*$ respectively. We observe that the decomposition of low-rank matrix $\mathbf{L}^*$ in Section 3.2 is not unique, since we have $\mathbf{L}^* = (\mathbf{Z}^*\mathbf{U})(\mathbf{Z}^*\mathbf{U})^\top$ for any $r \times r$ orthogonal matrix $\mathbf{U}$. Thus, we define the following solution set for $\mathbf{Z}$:

$$\mathcal{U} = \{\widetilde{\mathbf{Z}} \in \mathbb{R}^{d \times r} | \widetilde{\mathbf{Z}} = \mathbf{Z}^*\mathbf{U} \text{ for some } \mathbf{U} \in \mathbb{R}^{r \times r} \text{ with } \mathbf{U}\mathbf{U}^\top = \mathbf{U}^\top\mathbf{U} = \mathbf{I}_r\}. \tag{4.2}$$

Note that $\sigma_1(\widetilde{\mathbf{Z}}) = \sqrt{\sigma_{\max}}$ and $\sigma_r(\widetilde{\mathbf{Z}}) = \sqrt{\sigma_{\min}}$ for any $\widetilde{\mathbf{Z}} \in \mathcal{U}$.

To measure the closeness between our estimator for $\mathbf{Z}$ and the unknown parameter $\mathbf{Z}^*$, we use the following distance $d(\cdot, \cdot)$, which is invariant to rotation. Similar definition has been used in [45, 30, 40].

**Definition 4.3.** Define the distance between $\mathbf{Z}$ and $\mathbf{Z}^*$ as $d(\mathbf{Z}, \mathbf{Z}^*) = \min_{\widetilde{\mathbf{Z}} \in \mathcal{U}} \|\mathbf{Z} - \widetilde{\mathbf{Z}}\|_F$, where $\mathcal{U}$ is the solution set defined in (4.2).

At the core of our proof technique is the first-order stability condition on the population loss function. In detail, the population loss function is defined as the expectation of sample loss function in (3.3):

$$p(\mathbf{S}, \mathbf{L}) = \text{tr}(\boldsymbol{\Sigma}^*(\mathbf{S} + \mathbf{L})) - \log|\mathbf{S} + \mathbf{L}|. \tag{4.3}$$

For the ease of presentation, we define two balls around $\mathbf{S}^*$ and $\mathbf{Z}^*$ respectively: $\mathbb{B}_F(\mathbf{S}^*, R) = \{\mathbf{S} \in \mathbb{R}^{d \times d} : \|\mathbf{S} - \mathbf{S}^*\|_F \le R\}, \mathbb{B}_d(\mathbf{Z}^*, R) = \{\mathbf{Z} \in \mathbb{R}^{d \times r} : d(\mathbf{Z}, \mathbf{Z}^*) \le R\}$. Then the first-order stability condition is stated as follows.

**Condition 4.4** (First-order Stability)**.** Suppose $\mathbf{S} \in \mathbb{B}_F(\mathbf{S}^*, R), \mathbf{Z} \in \mathbb{B}_d(\mathbf{Z}^*, R)$ for some $R > 0$; by definition we have $\mathbf{L} = \mathbf{Z}\mathbf{Z}^\top$ and $\mathbf{L}^* = \mathbf{Z}^*\mathbf{Z}^{*\top}$. The gradient of population loss function with respect to $\mathbf{S}$ satisfies

$$\|\nabla_{\mathbf{S}} p(\mathbf{S}, \mathbf{L}) - \nabla_{\mathbf{S}} p(\mathbf{S}, \mathbf{L}^*)\|_F \le \gamma_2 \cdot \|\mathbf{L} - \mathbf{L}^*\|_F.$$

The gradient of the population loss function with respect to $\mathbf{L}$ satisfies

$$\|\nabla_{\mathbf{L}} p(\mathbf{S}, \mathbf{L}) - \nabla_{\mathbf{L}} p(\mathbf{S}^*, \mathbf{L})\|_F \le \gamma_1 \cdot \|\mathbf{S} - \mathbf{S}^*\|_F,$$

where $\gamma_1, \gamma_2 > 0$ are constants.

Condition 4.4 requires the population loss function has a variant of Lipschitz continuity for the gradient. Note that the gradient is taken with respect to one variable ($\mathbf{S}$ or $\mathbf{L}$), while the Lipschitz continuity is with respect to the other variable. Also, the Lipschitz property is defined only between the true parameters $\mathbf{S}^*, \mathbf{L}^*$ and arbitrary elements $\mathbf{S} \in \mathbb{B}_F(\mathbf{S}^*, R)$ and $\mathbf{L} = \mathbf{Z}\mathbf{Z}^\top$ such that $\mathbf{Z} \in \mathbb{B}_d(\mathbf{Z}^*, R)$. It should be noted that Condition 4.4, as is verified in the appendix, is inspired by a similar condition originally introduced in [2]. We extend it to the loss function of LVGMM with both sparse and low-rank structures, which plays an important role in the analysis.

The following theorem characterize the theoretical properties of Algorithm 1.

**Theorem 4.5.** Suppose Assumptions 4.1 and 4.2 hold. Assume that the sample size satisfies $n \geq 484\|\mathbf{\Omega}^*\|_{1,1}\nu^2 r s^* \log d/(25R^2\sigma_{\min})$ and the sparsity of the unknown sparse matrix satisfies $s^* \leq 25d^2 R^2\sigma_{\min}/(121r\alpha^{*2})$, where $R = \min\{1/4\sqrt{\sigma_{\max}}, 1/(2\nu), \sqrt{\sigma_{\min}}/(6.5\nu^2)\}$. Then with probability at least $1 - C/d$, the initial points $\widehat{\mathbf{S}}^{(0)}, \widehat{\mathbf{Z}}^{(0)}$ obtained by the *initialization* stage of Algorithm 1 satisfies

$$\left\|\widehat{\mathbf{S}}^{(0)} - \mathbf{S}^*\right\|_F \leq R, \quad \text{and} \quad d\left(\widehat{\mathbf{Z}}^{(0)}, \mathbf{Z}^*\right) \leq R, \tag{4.4}$$

where $C > 0$ is an absolute constant. Furthermore, suppose Condition 4.4 holds. Let the step sizes satisfy $\eta \leq C_0/(\sigma_{\max}\nu^2)$ and $\eta' \leq C_0\sigma_{\min}/(\sigma_{\max}\nu^4)$, and the sparsity parameter satisfies $s \geq (4(1/(2\sqrt{\rho}) - 1)^2 + 1)s^*$, where $C_0 > 0$ is a constant that can be chosen arbitrarily small. Let $\rho$ and $\tau$ be

$$\rho = \max\left\{1 - \frac{\eta}{\nu^2}, 1 - \frac{\eta'\sigma_{\min}}{\nu^2}\right\}, \qquad \tau = \max\left\{\frac{48C_0^2}{\sigma_{\max}^2\nu^4}\frac{s^* \log d}{n}, \frac{32C_0^2\sigma_{\min}^2}{\sigma_{\max}\nu^6}\frac{rd}{n}\right\}.$$

Then for any $t \geq 1$, with probability at least $1 - C_1/d$, the output of Algorithm 1 satisfies

$$\max\left\{\left\|\widehat{\mathbf{S}}^{(t+1)} - \mathbf{S}^*\right\|_F^2, d^2(\widehat{\mathbf{Z}}^{(t+1)}, \mathbf{Z}^*)\right\} \leq \underbrace{\frac{\tau}{1 - \sqrt{\rho}}}_{\text{statistical error}} + \underbrace{\sqrt{\rho^{t+1}} \cdot R}_{\text{optimization error}}, \tag{4.5}$$

where $C_1 > 0$ is an absolute constant.

In Theorem 4.5, $\rho$ is the contraction parameter of linear convergence rate, and it depends on the step size $\eta$. Therefore, we can always choose a sufficiently small step size by choosing a small enough $C_0$, such that $\rho$ is strictly between 0 and 1.

**Remark 4.6.** (4.4) suggests that, in order to ensure that the initial points returned by the initialization stage of Algorithm 1 fall in small neighborhoods of $\mathbf{S}^*$ and $\mathbf{Z}^*$, we require $n = O(s^* \log d)$, which essentially attains the optimal sample complexity for LVGGM estimation. In addition, we require $s^* \lesssim d^2/(r\alpha^{*2})$, which means the unknown sparse matrix cannot be too dense.

**Remark 4.7.** (4.5) suggests that the estimation error of the output of Algorithm 1 consists of two terms: the first term is the statistical error, and the second term is the optimization error. The statistical error comes from $\tau$ and scales as $\max\{O_p(\sqrt{s^* \log d/n}), O_p(\sqrt{rd/n})\}$, where $O_p(\sqrt{s^* \log d/n})$ corresponds to the statistical error of $\mathbf{S}^*$, and $O_p(\sqrt{rd/n})$ corresponds to the statistical error of $\mathbf{L}^*$ [1]. This matches the minimax optimal rate of estimation errors in Frobenius norm for LVGGM estimation [9, 1, 24]. For the optimization error, note that $\sigma_{\max}$ and $\sigma_{\min}$ are fixed constants. For a sufficiently small constant $C_0$, we can always ensure $\rho < 1$, and this establishes the linear convergence rate for Algorithm 1. Actually, after $T \geq \max\{O(\log(\nu^4 n/(s^* \log d))), O(\log(\nu^6 n/(rd)))\}$ iterations, the total estimation error of our algorithm achieves the same order as the statistical error.

**Remark 4.8.** Our statistical rate is sharp, because our theoretical analysis is conducted uniformly over the neighborhood of true parameters $\mathbf{S}^*$ and $\mathbf{Z}^*$, rather than doing sample splitting. This is another big advantage of our approach over existing algorithms which are also built upon first-order stability [2, 36] but rely on sample splitting technique.

# 5 Experiments

In this section, we present numerical results on both synthetic and real datasets to verify the theoretical properties of our algorithm, and compare it with the state-of-the-art methods. Specifically, we

compare our method, denoted by **AltGD**, with two convex relaxation based methods for estimating LVGGM: (1) LogdetPPA [9, 32] for solving log-determinant semidefinite programs, denoted by **PPA**, and (2) the alternating direction method of multipliers in [22, 24], denoted by **ADMM**. We also considered alternatives of the convex methods which use the randomized SVD method [15] in each iteration. However, the randomized SVD method still needs to compute a full SVD for nuclear norm regularization and in our experiments, we found that it is slower than the full SVD method implemented in [22]. Thus, we only report the results of the orignial convex relaxations in [9, 32, 22, 24]. The implementation of these two methods were downloaded from the authors' website. All numerical experiments were run in MATLAB R2015b on a laptop with Intel Core i5 2.7 GHz CPU and 8GB of RAM.

## 5.1 Synthetic Data

In the synthetic experiment, we first validate the performance of our method on the latent variable GGM. Then we show that our method also performs well on a more general GGM where the precision matrix is the sum of an arbitrary sparse matrix $\mathbf{S}^*$ and arbitrary low rank matrix $\mathbf{L}^*$. Specifically, we generated data according to the following two schemes:

- Scheme I: we generated data from the latent variable GGM defined in Section 3.1. In detail, the dimension of observed data is $d$ and the number of latent variables is $r$. We randomly generated a sparse positive definite matrix $\widetilde{\mathbf{\Omega}} \in \mathbb{R}^{(d+r)\times(d+r)}$, with sparsity $s^* = 0.02d^2$. According to (3.1), the sparse component of the precision matrix is $\mathbf{S}^* := \widetilde{\mathbf{\Omega}}_{1:d;1:d}$ and the low-rank component is $\mathbf{L}^* := -\widetilde{\mathbf{\Omega}}_{1:d;(d+1):(d+r)}[\widetilde{\mathbf{\Omega}}_{(d+1):(d+r);(d+1):(d+r)}]^{-1}\widetilde{\mathbf{\Omega}}_{(d+1):(d+r);1:d}$. Then we sampled data $\mathbf{X}_1,\ldots,\mathbf{X}_n$ from distribution $N(\mathbf{0},(\mathbf{\Omega}^*)^{-1})$, where $\mathbf{\Omega}^* = \mathbf{S}^* + \mathbf{L}^*$ is the true precision matrix.
- Scheme II: the dimension of observed data is $d$ and the number of latent variables is $r$. $\mathbf{S}^*$ is a symmetric positive definite matrix with entries randomly generated from $[-1,1]$ with sparsity $s^* = 0.05d^2$. $\mathbf{L}^* = \mathbf{Z}^*\mathbf{Z}^{*\top}$, where $\mathbf{Z}^* \in \mathbb{R}^{d\times r}$ with entries randomly generated from $[-1,1]$. Then we sampled data $\mathbf{X}_1,\ldots,\mathbf{X}_n$ from multivariate normal distribution $N(\mathbf{0},(\mathbf{\Omega}^*)^{-1})$ with $\mathbf{\Omega}^* = \mathbf{S}^* + \mathbf{L}^*$ being the true precision matrix.

Table 1: Scheme I: estimation errors of sparse and low-rank components $\mathbf{S}^*$ and $\mathbf{L}^*$ as well as the true precision matrix $\mathbf{\Omega}^*$ in terms of Frobenius norm on different synthetic datasets. Data were generated from LVGGM and results were reported on 10 replicates in each setting.

| Setting | Method | $\|\widehat{\mathbf{S}}^{(T)} - \mathbf{S}^*\|_F$ | $\|\widehat{\mathbf{L}}^{(T)} - \mathbf{L}^*\|_F$ | $\|\widehat{\mathbf{\Omega}}^{(T)} - \mathbf{\Omega}^*\|_F$ | Time $(s)$ |
|---|---|---|---|---|---|
| $d=100, r=2, n=2000$ | PPA | 0.7335±0.0352 | 0.0170±0.0125 | 0.7350±0.0359 | 1.1610 |
| | ADMM | 0.7521±0.0288 | 0.0224±0.0115 | 0.7563±0.0298 | 1.1120 |
| | AltGD | 0.6241±0.0668 | 0.0113±0.0014 | 0.6236±0.0669 | 0.0250 |
| $d=500, r=5, n=10000$ | PPA | 0.9803±0.0192 | 0.0195±0.0046 | 0.9813±0.0192 | 35.7220 |
| | ADMM | 1.0571±0.0135 | 0.0294±0.0041 | 1.0610±0.0134 | 25.8010 |
| | AltGD | 0.8212±0.0143 | 0.0125±0.0000 | 0.8210±0.0143 | 0.4800 |
| $d=1000, r=8, n=2.5\times10^4$ | PPA | 1.1620±0.0177 | 0.0224±0.0034 | 1.1639±0.0179 | 356.7360 |
| | ADMM | 1.1867±0.0253 | 0.0356±0.0033 | 1.1869±0.0254 | 156.5550 |
| | AltGD | 0.9016±0.0245 | 0.0167±0.0030 | 0.9021±0.0244 | 7.4740 |
| $d=5000, r=10, n=2\times10^5$ | PPA | 1.4822±0.0302 | 0.0371±0.0052 | 1.4824±0.0120 | 33522.0200 |
| | ADMM | 1.5010±0.0240 | 0.0442±0.0068 | 1.5012±0.0240 | 21090.7900 |
| | AltGD | 1.3449±0.0073 | 0.0208±0.0014 | 1.3449±0.0084 | 445.6730 |

In both schemes, we conducted experiments in different settings of $d, n, s^*$ and $r$. The step sizes of our method were set as $\eta = c_1/(\sigma_{\max}\nu^2)$ and $\eta' = c_1\sigma_{\min}/(\sigma_{\max}\nu^4)$ according to Theorem 4.5, where $c_1 = 0.25$. The thresholding parameter $s$ is set as $c_2 s^*$, where $c_2 > 1$ was selected by 4-fold cross-validation. The regularization parameters for $\ell_{1,1}$-norm and nuclear norm in **PPA** and **ADMM** and the tuning parameter $r$ in our algorithm were selected by 4-fold cross-validation. Under both schemes, we repeatedly generated 10 datasets for each setting of $d, n, s^*$ and $r^*$, and calculated the mean and standard error of the estimation error. We summarize the results of Scheme I over 10 replications in Table 1. Note that our algorithm **AltGD** outputs a slightly better estimator in terms of estimation errors compared with **PPA** and **ADMM**. It should also be noted that they do not differ too much because their statistical rates should be in the same order. To demonstrate the efficiency of our algorithm, we also reported the mean CPU time in the last column of Table 1. We observe

significant speed-ups brought by our algorithm, which is almost 50 times faster than the convex ones. In particular, when the dimension $d$ scales up to several thousands, the computation of SVD in **PPA** and **ADMM** takes enormous time and therefore the computational time of them increases dramatically. We report the averaged results of Scheme II over 10 repetitions in Table 2. Again, it can be seen that our method **AltGD** achieves comparable or slightly better estimators in terms of estimation errors in Frobenius norm compared against **PPA** and **ADMM**. Our method **AltGD** is nearly 50 times faster than the other two methods based on convex algorithms.

Table 2: Scheme II: estimation errors of sparse and low-rank components $\mathbf{S}^*$ and $\mathbf{L}^*$ as well as the true precision matrix $\mathbf{\Omega}^*$ in terms of Frobenius norm on different synthetic datasets. Data were generated from multivariate distribution where the precision matrix is the sum of an arbitrary sparse matrix and an arbitrary low-rank matrix. Results were reported on 10 replicates in each setting.

| Setting | Method | $\|\widehat{\mathbf{S}}^{(T)} - \mathbf{S}^*\|_F$ | $\|\widehat{\mathbf{L}}^{(T)} - \mathbf{L}^*\|_F$ | $\|\widehat{\mathbf{\Omega}}^{(T)} - \mathbf{\Omega}^*\|_F$ | Time $(s)$ |
|---|---|---|---|---|---|
| $d=100, r=2, n=$ 2000 | PPA | 0.5710±0.0319 | 0.6231±0.0261 | 0.8912±0.0356 | 1.6710 |
| | ADMM | 0.6198±0.0361 | 0.5286±0.0308 | 0.8588±0.0375 | 1.2790 |
| | AltGD | 0.5639±0.0905 | 0.4824±0.0323 | 0.7483±0.0742 | 0.0460 |
| $d=500, r=5, n=$ 10000 | PPA | 0.8140±0.0157 | 0.7802±0.0104 | 1.1363±0.0131 | 43.8000 |
| | ADMM | 0.8140±0.0157 | 0.7803±0.0104 | 1.1363±0.0131 | 25.8980 |
| | AltGD | 0.6139±0.0198 | 0.7594±0.0111 | 0.9718±0.0146 | 0.8690 |
| $d=1000, r=8, n=$ $2.5 \times 10^4$ | PPA | 0.9235±0.0193 | 1.1985±0.0084 | 1.4913±0.0162 | 487.4900 |
| | ADMM | 0.9209±0.0212 | 1.2131±0.0084 | 1.4975±0.0171 | 163.9350 |
| | AltGD | 0.7249±0.0158 | 0.9651±0.0093 | 1.2029±0.0141 | 7.1360 |
| $d=5000, r=10, n=$ $2 \times 10^5$ | PPA | 1.1883±0.0091 | 1.0970±0.0022 | 1.3841±0.0083 | 44098.6710 |
| | ADMM | 1.2846±0.0089 | 1.1568±0.0023 | 1.5324±0.0085 | 20393.3650 |
| | AltGD | 1.0681±0.0034 | 1.0685±0.0023 | 1.2068±0.0032 | 287.8630 |

In addition, we illustrate the convergence rate of our algorithm in Figure 1(a) and 1(b), where the x-axis is iteration number and y-axis is the estimation errors in Frobenius norm. We can see that our algorithm converges in dozens of iterations, which confirms our theoretical guarantee on linear convergence rate. We plot the overall estimation errors against the scaled statistical errors of $\mathbf{S}^{(T)}$ and $\mathbf{L}^{(T)}$ under different configurations of $d, n, s^*$ and $r$ in Figure 1(c) and 1(d). According to Theorem 4.5, $\|\widehat{\mathbf{S}}^{(t)} - \mathbf{S}^*\|_F$ and $\|\widehat{\mathbf{L}}^{(t)} - \mathbf{L}^*\|_F$ will converge to the statistical errors as the number of iterations $t$ goes up, which are in the order of $O(\sqrt{s^* \log d/n})$ and $O(\sqrt{rd/n})$ respectively. We can see that the estimation errors grow linearly with the theoretical rate, which validates our theoretical guarantee on the minimax optimal statistical rate.

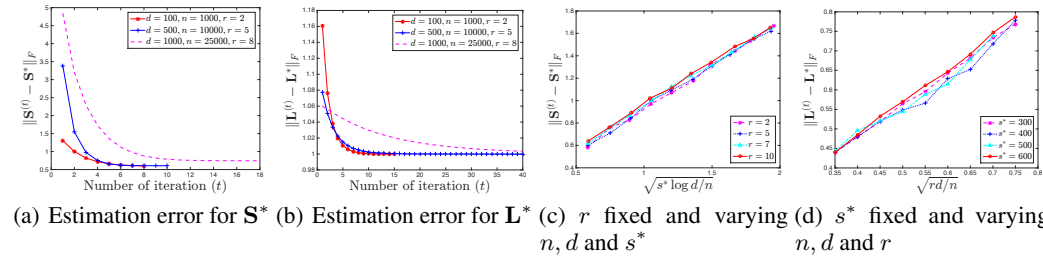

(a) Estimation error for $\mathbf{S}^*$ (b) Estimation error for $\mathbf{L}^*$ (c) $r$ fixed and varying $n, d$ and $s^*$ (d) $s^*$ fixed and varying $n, d$ and $r$

Figure 1: (a)-(b): Evolution of estimation errors with number of iterations $t$ going up with the sparsity parameter $s^*$ set as $0.02 \times d^2$ and varying $d, n$ and $r$. (c)-(d): Estimation errors $\|\widehat{\mathbf{S}}^{(T)} - \mathbf{S}^*\|_F$ and $\|\widehat{\mathbf{L}}^{(T)} - \mathbf{L}^*\|_F$ versus scaled statistical errors $\sqrt{s^* \log d/n}$ and $\sqrt{rd/n}$.

## 5.2 Genomic Data

In this subsection, we apply our method to TCGA breast cancer gene expression data to infer regulatory network. We downloaded the gene expression data from cBioPortal[2]. Here we focused on 299 breast cancer related transcription factors (TFs) and estimated the regulatory relationships for each pair of TFs over two breast cancer subtypes: luminal and basal. We compared our method **AltGD**

with **ADMM** and **PPA** which are all based on LVGGM. We also compared it with the graphical Lasso (**GLasso**) which only considers the sparse structure of precision matrix and ignores the latent variables; we chose QUIC[3] to solve the GLasso estimator. Regarding the benchmark standard, we used the "regulatory potential scores" between a pair of genes (a TF and a target gene) for these two breast cancer subtypes compiled based on both co-expression and TF ChIP-seq binding data from the Cistrome Cancer Database[4].

Table 3: Summary of CPU time of different methods on luminal subtype breast cancer dataset.

| Method | GLasso | PPA | ADMM | AltGD |
|---|---|---|---|---|
| Time ($s$) | 38.6310 | 85.0100 | 7.6700 | 0.1500 |

For luminal subtype, there are $n = 601$ samples and $d = 299$ TFs. The regularization parameters for $\ell_{1,1}$ norm in **GLasso**, for $\ell_{1,1}$ norm and nuclear norm in **PPA** and **ADMM** were tuned by grid search. The step sizes of **AltGD** were set as $\eta = 0.1/\widehat{\nu}^2$ and $\eta' = 0.1/\widehat{\nu}^4$, where $\widehat{\nu}$ is the maximal eigenvalue of sample covariance matrix. The thresholding parameter $s$ and number of latent variables $r$ were tuned by grid search. In Table 3, we present the CPU time of each method. Importantly, we can see that **AltGD** is the fastest among all the methods and is even more than 50 times faster than the second fastest method **ADMM**.

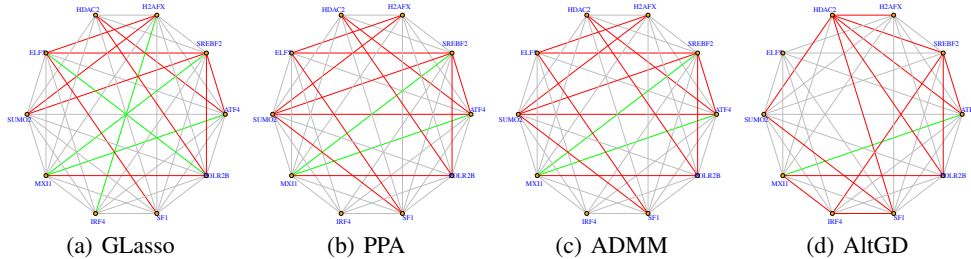

| (a) GLasso | (b) PPA | (c) ADMM | (d) AltGD |

Figure 2: An example of subnetwork in the transcriptional regulatory network of luminal breast cancer. Here gray edges are the interactions from the Cistrome Cancer Database; red edges are the ones inferred by the respective methods; green edges are incorrectly inferred interactions.

To demonstrate the performances of different methods on recovering the overall transcriptional regulatory network, we randomly selected 10 TFs in the benchmark network and plotted the subnetwork in Figure 2 which has 70 edges with nonzero regulatory potential scores. Specifically, the gray edges form the benchmark network, the red edges are those identified correctly and the green edges are those incorrectly inferred by each method. We can observe from Figure 2 that the methods based on LVGGMs are able to recover more edges accurately than graphical Lasso because of the intervention of latent variables. We remark that all the methods were not able to completely recover the entire regulatory network partly because we only used the gene expression data for TFs (instead of all genes) and the regulatory potential scores from the Cistome Cancer Database also used TF binding information. Due to space limit, we postpone additional experimental results to the appendix.

## 6   Conclusions

In this paper, to speed up the learning of LVGGM, we proposed a sparsity constrained maximum likelihood estimator based on matrix factorization. We developed an efficient alternating gradient descent algorithm, and proved that the proposed algorithm is guaranteed to converge to the unknown sparse and low-rank matrices with a linear convergence rate up to the optimal statical error. Experiments on both synthetic and real world genomic data supported our theory.

## Acknowledgements

We would like to thank the anonymous reviewers for their helpful comments. This research was sponsored in part by the National Science Foundation IIS-1652539, IIS-1717205 and IIS-1717206. The views and conclusions contained in this paper are those of the authors and should not be interpreted as representing any funding agencies.

## Footnotes

[1]While the derived error bound in (4.5) is for $\widehat{\mathbf{Z}}^{(t)}$, it is in the same order as the error bound of $\widehat{\mathbf{L}}^{(t)}$ by definition.

[2] http://www.cbioportal.org/

[3]`http://www.cs.utexas.edu/~sustik/QUIC/`

[4]`http://cistrome.org/CistromeCancer/`

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
