[Supplementary Material]

# A Proof of Main Theoretical Results

In this section, we prove our main theories. The proof of Theorem 4.5 requires the following two lemmas.

**Lemma A.1.** Suppose that Assumptions 4.1 and 4.2 hold. Choose the thresholding parameter as $s > s^*$. Then with probability at least $1 - C'/d$, the initial points $\widehat{\mathbf{S}}^{(0)}, \widehat{\mathbf{Z}}^{(0)}$ obtained by initialization stage in Algorithm 1 satisfy

$$\left\|\widehat{\mathbf{S}}^{(0)} - \mathbf{S}^*\right\|_F \leq \sqrt{s^* + s}\left(\frac{\alpha^*}{d} + 2\|\mathbf{\Omega}^*\|_{1,1}\nu\sqrt{\frac{\log d}{n}}\right),$$

$$d(\widehat{\mathbf{Z}}^{(0)}, \mathbf{Z}^*) \leq \frac{11\alpha^*\sqrt{r(s^* + s)}}{5d\sqrt{\sigma_{\min}}} + \frac{22\nu^3}{5\sqrt{\sigma_{\min}}}\sqrt{\frac{rd}{n}} + \frac{22\|\mathbf{\Omega}^*\|_{1,1}\nu}{5\sqrt{\sigma_{\min}}}\sqrt{\frac{r(s^* + s)\log d}{n}},$$

where $C' > 0$ is an absolute constant.

Lemma A.1 indicates that under certain conditions the initial points obtained by *initialization* stage of Algorithm 1 are sufficiently close to $\mathbf{S}^*$ and $\mathbf{Z}^*$ respectively.

**Lemma A.2.** Suppose Assumption 4.1 and Condition 4.4 hold. Suppose the initial solutions satisfy $\widehat{\mathbf{S}}^{(0)} \in \mathbb{B}_F(\mathbf{S}^*, R)$ and $\widehat{\mathbf{Z}}^{(0)} \in \mathbb{B}_d(\mathbf{Z}^*, R)$, where $R = \min\{1/4\sqrt{\sigma_{\max}}, 1/(2\nu), \sqrt{\sigma_{\min}}/(6.5\nu^2)\}$. In Algorithm 1, let the step sizes satisfy $\eta \leq C_0/(\sigma_{\max}\nu^2)$ and $\eta' \leq C_0\sigma_{\min}/(\sigma_{\max}\nu^4)$, and the sparsity parameter satisfies $s \geq \left(4(1/(2\sqrt{\rho}) - 1)^2 + 1\right)s^*$, where $C_0 > 0$ is a sufficiently small constant. Let $\rho$ and $\tau$ be

$$\rho = \max\left\{1 - \frac{\eta}{\nu^2}, 1 - \frac{\eta'\sigma_{\min}}{\nu^2}\right\}, \qquad \tau = \max\left\{\frac{48C_0^2}{\sigma_{\max}^2\nu^4}\frac{s^*\log d}{n}, \frac{32C_0^2\sigma_{\min}^2}{\sigma_{\max}\nu^6}\frac{rd}{n}\right\}.$$

Then for any $t \geq 1$, with probability at least $1 - C_1/d$, the output of Algorithm 1 satisfies

$$\max\left\{\left\|\widehat{\mathbf{S}}^{(t+1)} - \mathbf{S}^*\right\|_F^2, d^2(\widehat{\mathbf{Z}}^{(t+1)}, \mathbf{Z}^*)\right\} \leq \frac{\tau}{1 - \sqrt{\rho}} + \sqrt{\rho^{t+1}} \cdot R, \qquad \text{(A.1)}$$

where $C_1 > 0$ is an absolute constant.

Lemma A.2 suggests that the estimation error consists of two terms: the first term is the statistical error, and the second term is the optimization error of our algorithm.

*Proof of Theorem 4.5.* The proof of Theorem 4.5 follows from combining Lemma A.1 and Lemma A.2. We only need to derive the conditions on the sample size $n$ and sparsity of $\mathbf{S}^*$. Specifically, for $R$ defined as in Lemma A.2, we need

$$\sqrt{s^* + s}\left(\frac{\alpha^*}{d} + 2\|\mathbf{\Omega}^*\|_{1,1}\nu\sqrt{\frac{\log d}{n}}\right) \leq R,$$

$$\frac{11\alpha^*\sqrt{r(s^* + s)}}{5d\sqrt{\sigma_{\min}}} + \frac{22\nu^3}{5\sqrt{\sigma_{\min}}}\sqrt{\frac{rd}{n}} + \frac{22\|\mathbf{\Omega}^*\|_{1,1}\nu}{5\sqrt{\sigma_{\min}}}\sqrt{\frac{r(s^* + s)\log d}{n}} \leq R,$$

to ensure the initial points obtained by the initialization stage of Algorithm 1 lie in balls of unknown matrices with radius $R$. Simple calculation yields the condition on the sample size $n$ and sparsity $s^*$ as follows:

$$n \geq 484\|\mathbf{\Omega}^*\|_{1,1}\nu^2rs^*\log d/(25R^2\sigma_{\min}), \quad \text{and} \quad s^* \leq 25d^2R^2\sigma_{\min}/(121r\alpha^{*2}).$$

This completes the proof. □

# B Proof of Technical Lemmas in Appendix A

In this section, we provide the proofs for the technical lemma used in the proof of our main theory.

## B.1 Proof of Lemma A.1

Now we prove that our initial points $\widehat{\mathbf{S}}^{(0)}$ and $\widehat{\mathbf{L}}^{(0)}$ in Algorithm 1 lie in small neighborhoods of $\mathbf{S}^*$ and $\mathbf{Z}^*$. Note that our analysis of the initialization is inspired by the proof of Theorem 1 in [40], and extends that to the noisy case. We first lay out the following lemma, which is useful in our proof.

**Lemma B.1.** For any symmetric matrix $\mathbf{A} \in \mathbb{R}^{d \times d}$ with $\|\mathbf{A}\|_{0,0} = s_0$, we have

$$\|\mathbf{A}\|_2 \leq \sqrt{s_0}\|\mathbf{A}\|_{\infty,\infty}.$$

*Proof of Lemma A.1.* Let $\mathbf{E} = \boldsymbol{\Omega}^* - \widehat{\boldsymbol{\Sigma}}^{-1} = \mathbf{S}^* + \mathbf{L}^* - \widehat{\boldsymbol{\Sigma}}^{-1}$, where $\widehat{\boldsymbol{\Sigma}} = 1/n \sum_{i=1}^{n} \mathbf{X}_i \mathbf{X}_i^\top$ is the sample covariance matrix. According to Algorithm 1, we have $\widehat{\mathbf{S}}^{(0)} = \mathcal{HT}_s(\widehat{\boldsymbol{\Sigma}}^{-1})$. We define $\mathbf{Y} = \boldsymbol{\Omega}^* - \widehat{\mathbf{S}}^{(0)} = \mathbf{E} + \widehat{\boldsymbol{\Sigma}}^{-1} - \widehat{\mathbf{S}}^{(0)}$, which immediately implies that $\mathbf{Y} - \mathbf{L}^* = \mathbf{S}^* - \widehat{\mathbf{S}}^{(0)}$ and that $\mathrm{supp}(\mathbf{Y} - \mathbf{L}^*) = \mathrm{supp}(\widehat{\mathbf{S}}^{(0)}) \cup \mathrm{supp}(\mathbf{S}^*)$. Specifically,

- For $(j,k) \in \mathrm{supp}(\widehat{\mathbf{S}}^{(0)})$, we have $[\mathbf{Y} - \mathbf{L}^*]_{jk} = [\mathbf{E} - \mathbf{L}^*]_{jk}$, since $[\widehat{\boldsymbol{\Sigma}}^{-1} - \widehat{\mathbf{S}}^{(0)}]_{jk} = 0$ by thresholding.

- For $(j,k) \in \mathrm{supp}(\mathbf{S}^*)/\mathrm{supp}(\widehat{\mathbf{S}}^{(0)})$, we have $|[\mathbf{Y} - \mathbf{L}^*]_{jk}| = |S_{jk}^*| \leq 2\|\mathbf{L}^*\|_{\infty,\infty} + \|\mathbf{E}\|_{\infty,\infty}$. Otherwise $|[\widehat{\boldsymbol{\Sigma}}^{-1}]_{jk}| = |[\mathbf{S}^* + \mathbf{L}^* - \mathbf{E}]_{jk}| \geq |S_{jk}^*| - |[\mathbf{L}^* - \mathbf{E}]_{jk}| \geq \|\mathbf{L}^*\|_{\infty,\infty}$. Since $\|\mathbf{S}^*\|_{0,0} \leq s^*$ and $s \geq s^*$, this means that $|[\widehat{\boldsymbol{\Sigma}}^{-1}]_{jk}|$ is greater than at least $d - s^* \geq d - s$ entries in $\widehat{\boldsymbol{\Sigma}}^{-1}$, which immediately yields that $(j,k) \in \mathrm{supp}(\widehat{\mathbf{S}}^{(0)})$. This contradiction leads to our claim that $|[\mathbf{Y} - \mathbf{L}^*]_{jk}| = |S_{jk}^*| \leq 2\|\mathbf{L}^*\|_{\infty,\infty} + \|\mathbf{E}\|_{\infty,\infty}$.

Thus, we've proved that

$$\|\mathbf{Y} - \mathbf{L}^*\|_{\infty,\infty} \leq 2\|\mathbf{L}^*\|_{\infty,\infty} + \|\mathbf{E}\|_{\infty,\infty}. \tag{B.1}$$

For $\mathbf{L}^* = \mathbf{V}^* D^* \mathbf{V}^{*\top}$, by spikiness condition of $\mathbf{L}^*$ in Assumption 4.2, we have

$$\|\mathbf{L}^*\|_{\infty,\infty} \leq \frac{\alpha^*}{d}. \tag{B.2}$$

Moreover, since $\mathbf{E} = \boldsymbol{\Sigma}^{*-1} - \widehat{\boldsymbol{\Sigma}}^{-1}$, we notice that

$$\|\boldsymbol{\Sigma}^{*-1} - \widehat{\boldsymbol{\Sigma}}^{-1}\|_{\infty,\infty} = \|\widehat{\boldsymbol{\Sigma}}^{-1}(\widehat{\boldsymbol{\Sigma}} - \boldsymbol{\Sigma}^*)\boldsymbol{\Sigma}^{*-1}\|_{\infty,\infty} \leq 2\|\boldsymbol{\Omega}^*\|_{1,1}\nu\sqrt{\frac{\log d}{n}} \tag{B.3}$$

holds with probability at least $1 - C/d$, where the last inequality is due to Lemma E.2. Combining (B.1), (B.2) and (B.3), it finally yields that

$$\left\|\widehat{\mathbf{S}}^{(0)} - \mathbf{S}^*\right\|_F \leq \sqrt{s^* + s}\|\mathbf{Y} - \mathbf{L}^*\|_{\infty,\infty} \leq \sqrt{s^* + s}\left(\frac{\alpha^*}{d} + 2\|\boldsymbol{\Omega}^*\|_{1,1}\nu\sqrt{\frac{\log d}{n}}\right)$$

holds with probability at least $1 - C/d$. It follows from Lemma B.1 that

$$\|\mathbf{Y} - \mathbf{L}^*\|_2 \leq \sqrt{s^* + s}\|\mathbf{Y} - \mathbf{L}^*\|_{\infty,\infty} \leq 2\sqrt{s^* + s}\|\mathbf{L}^*\|_{\infty,\infty} + \sqrt{s^* + s}\|\mathbf{E}\|_{\infty,\infty} \tag{B.4}$$

Since $\widehat{\mathbf{Z}}^{(0)}\widehat{\mathbf{Z}}^{(0)\top}$ is the rank $r$ approximation of $\widehat{\boldsymbol{\Sigma}}^{-1} - \widehat{\mathbf{S}}^{(0)} = \mathbf{Y} - \mathbf{E}$, we have

$$\|\widehat{\mathbf{Z}}^{(0)}\widehat{\mathbf{Z}}^{(0)\top} - (\mathbf{Y} - \mathbf{E})\|_2 = \sigma_{r+1}(\mathbf{Y} - \mathbf{E}).$$

Noting that $\sigma_{r+1}(\mathbf{L}^*) = 0$, applying Weyl's theorem yields

$$|\sigma_{r+1}(\mathbf{Y} - \mathbf{E}) - \sigma_{r+1}(\mathbf{L}^*)| \leq \|(\mathbf{Y} - \mathbf{E}) - \mathbf{L}^*\|_2,$$

which immediately implies

$$\|\widehat{\mathbf{Z}}^{(0)}\widehat{\mathbf{Z}}^{(0)\top} - \mathbf{L}^*\|_2 \leq \|\widehat{\mathbf{Z}}^{(0)}\widehat{\mathbf{Z}}^{(0)\top} - (\mathbf{Y} - \mathbf{E})\|_2 + \|\mathbf{Y} - \mathbf{E} - \mathbf{L}^*\|_2$$
$$\leq 2\|\mathbf{Y} - \mathbf{E} - \mathbf{L}^*\|_2.$$

Thus submitting (B.2), (B.4) and Lemma E.3 into the above inequality, we obtain

$$
\begin{aligned}
\big\|\widehat{\mathbf{Z}}^{(0)}\widehat{\mathbf{Z}}^{(0)\top} - \mathbf{L}^*\big\|_F &\leq 2\sqrt{r}\big(\|\mathbf{Y} - \mathbf{L}^*\|_2 + \|\mathbf{E}\|_2\big) \\
&\leq 2\sqrt{r(s^*+s)}\|\mathbf{L}^*\|_{\infty,\infty} + 2\sqrt{r(s^*+s)}\|\mathbf{E}\|_{\infty,\infty} + 2\sqrt{r}\|\mathbf{E}\|_2 \\
&\leq \frac{2\alpha^*\sqrt{r(s^*+s)}}{d} + 4\|\mathbf{\Omega}^*\|_{1,1}\nu\sqrt{\frac{r(s^*+s)\log d}{n}} + 4\nu^3\sqrt{\frac{rd}{n}}, \quad \text{(B.5)}
\end{aligned}
$$

with probability at least $1 - C'/d$, where $C' > 0$ is an absolute constant. And by Lemma E.4 we further get

$$
d(\widehat{\mathbf{Z}}^{(0)}, \mathbf{Z}^*) \leq \frac{11\alpha^*\sqrt{r(s^*+s)}}{5\sqrt{\sigma_{\min}}d} + \frac{22\|\mathbf{\Omega}^*\|_{1,1}\nu}{5\sqrt{\sigma_{\min}}}\sqrt{\frac{r(s^*+s)\log d}{n}} + \frac{22\nu^3}{5\sqrt{\sigma_{\min}}}\sqrt{\frac{rd}{n}}, \quad \text{(B.6)}
$$

with probability at least $1 - C'/d$, which completes the proof. $\qquad\square$

## B.2  Proof of Lemma A.2

For simplicity of the proof, we introduce the following notations that give the gradient descent updating based on the population objective function

$$
\begin{aligned}
\mathbf{S}^{(t+0.5)} &= \widehat{\mathbf{S}}^{(t)} - \eta\nabla_{\mathbf{S}}q\big(\widehat{\mathbf{S}}^{(t)}, \widehat{\mathbf{Z}}^{(t)}\big), \\
\mathbf{Z}^{(t+1)} &= \widehat{\mathbf{Z}}^{(t)} - \eta'\nabla_{\mathbf{Z}}q\big(\widehat{\mathbf{S}}^{(t)}, \widehat{\mathbf{Z}}^{(t)}\big),
\end{aligned}
\quad \text{(B.7)}
$$

where the population objective function $q(\mathbf{S}, \mathbf{Z}) = \mathbb{E}[q_n(\mathbf{S}, \mathbf{Z})]$ and $q_n(\mathbf{S}, \mathbf{Z})$ is defined in (3.4). Here $\mathbf{S}^{(t+0.5)}$ and $\mathbf{Z}^{(t+1)}$ are the population version of $\widehat{\mathbf{S}}^{(t+0.5)}$ and $\widehat{\mathbf{Z}}^{(t+1)}$ in Algorithm 1. In order to prove our main theorem, we layout some useful lemmas here first.

**Lemma B.2.** Let $\mathbf{S}^{(t+0.5)} = \widehat{\mathbf{S}}^{(t)} - \eta\nabla_{\mathbf{S}}q\big(\widehat{\mathbf{S}}^{(t)}, \widehat{\mathbf{Z}}^{(t)}\big)$ be the population version of $\widehat{\mathbf{S}}^{(t+0.5)}$. For the gradient descent updating of $\mathbf{S}$, if step size satisfies $\eta \leq 1/(L+\mu)$, then we have

$$
\big\|\mathbf{S}^{(t+0.5)} - \mathbf{S}^*\big\|_F^2 \leq \left(1 - \frac{2\eta\mu L}{L+\mu}\right)\|\widehat{\mathbf{S}}^{(t)} - \mathbf{S}^*\|_F^2 + \frac{25\eta^2\gamma_2^2\sigma_{\max}}{8}d^2(\widehat{\mathbf{Z}}^{(t)}, \mathbf{Z}^*),
$$

where $L = 4\nu^2, \mu = 1/(4\nu^2)$ and $\gamma_2 = 8\nu^2$.

And the corresponding result for $\mathbf{Z}$:

**Lemma B.3.** Let $\mathbf{Z}^{(t+1)} = \widehat{\mathbf{Z}}^{(t)} - \eta'\nabla_{\mathbf{Z}}q\big(\widehat{\mathbf{S}}^{(t)}, \widehat{\mathbf{Z}}^{(t)}\big)$ be the population version of $\widehat{\mathbf{Z}}^{(t+1)}$. The gradient descent algorithm of $\mathbf{Z}$ with step size $\eta' \leq 1/[16(L+\mu)\sigma_{\max}]$ satisfies

$$
d^2\big(\mathbf{Z}^{(t+1)}, \mathbf{Z}^*\big) \leq \left(1 - \frac{\eta'\sigma_{\min}\mu L}{2(L+\mu)}\right)d^2(\widehat{\mathbf{Z}}^{(t)}, \mathbf{Z}^*) + \frac{25\eta'^2\gamma_1^2\sigma_{\max}}{8}\|\widehat{\mathbf{S}}^{(t)} - \mathbf{S}^*\|_F^2,
$$

where $L = 4\nu^2, \mu = 1/(4\nu^2)$, and $\gamma_1 = 8\nu^2$. $d(\cdot, \cdot)$ is the distance defined in Definition 4.3.

The following lemma serves similarly as a non-expansive property for hard thresholding operators, which is proved in Lemma 4.1 by [21].

**Lemma B.4** ([21]). $\boldsymbol{\theta}^* \in \mathbb{R}^d$ is a sparse vector with $\|\boldsymbol{\theta}\|_0 = s^*$. For any $\boldsymbol{\theta} \in \mathbb{R}^d$, let $\mathcal{HT}_s(\cdot)$ be the hard thresholding function which preserves the $s$ largest magnitudes. Then we have

$$
\|\mathcal{HT}_s(\boldsymbol{\theta}) - \boldsymbol{\theta}^*\|_2^2 \leq \left(1 + \frac{2\sqrt{s^*}}{\sqrt{s - s^*}}\right)\|\boldsymbol{\theta} - \boldsymbol{\theta}^*\|_2^2.
$$

The following lemma gives the statistical error of our model.

**Lemma B.5.** For a given sample with size $n$ and dimension $d$, we use $\epsilon_1(n,d)$ and $\epsilon_2(n,d)$ to denote the statistical errors. More specifically, uniformly for all $\mathbf{S}$ over ball $\mathbb{B}_F(\mathbf{S}^*, R)$, $\mathbf{Z}$ over ball $\mathbb{B}_d(\mathbf{Z}^*, R)$ we have that

$$
\|\nabla_{\mathbf{S}}q_n(\mathbf{S}, \mathbf{Z}) - \nabla_{\mathbf{S}}q(\mathbf{S}, \mathbf{Z})\|_{\infty,\infty} \leq \epsilon_1(n,d) = 2\sqrt{\frac{\log d}{n}}
$$

holds with probability at least $1 - C/d$. And

$$
\|\nabla_{\mathbf{Z}}q_n(\mathbf{S}, \mathbf{Z}) - \nabla_{\mathbf{Z}}q(\mathbf{S}, \mathbf{Z})\|_F \leq \epsilon_2(n,d) = 4\nu\sqrt{\sigma_{\max}}\sqrt{\frac{rd}{n}}
$$

holds with probability at least $1 - C'/d$.

The above lemma states that the differences between the gradients of the population and sample loss functions with respect to $\mathbf{S}$ and $\mathbf{Z}$ are bounded in terms of different matrix norms. It is pivotal to characterize the statistical error of the estimator from our algorithm.

Now we are going to prove the main theorem.

*Proof of Lemma A.2.* We show $\widehat{\mathbf{S}}^{(t)} \in \mathbb{B}_F(\mathbf{S}^*, R), \widehat{\mathbf{Z}}^{(t)} \in \mathbb{B}_d(\mathbf{Z}^*, R)$, for all $t = 0, 1, \ldots$ by mathematical induction. We already know the initial points $\widehat{\mathbf{S}}^{(0)} \in \mathbb{B}_F(\mathbf{S}^*, R)$ and $\widehat{\mathbf{Z}}^{(0)} \in \mathbb{B}_d(\mathbf{Z}^*, R)$ by the initialization stage in Algorithm 1. Next, suppose that we have $\widehat{\mathbf{S}}^{(t)} \in \mathbb{B}_F(\mathbf{S}^*, R)$, $\widehat{\mathbf{Z}}^{(t)} \in \mathbb{B}_d(\mathbf{Z}^*, R)$ and we want to show this holds for iteration $t + 1$ too.

Define $\mathcal{S}^* = \text{supp}(\mathbf{S}^*)$, $\mathcal{S}^{(t)} = \text{supp}(\widehat{\mathbf{S}}^{(t)})$, $\mathcal{S}^{(t+1)} = \text{supp}(\widehat{\mathbf{S}}^{(t+1)})$ and $\bar{\mathcal{S}} = \mathcal{S}^* \cup \mathcal{S}^{(t)} \cup \mathcal{S}^{(t+1)}$. Recall that $\widehat{\mathbf{S}}^{(t+0.5)} = \widehat{\mathbf{S}}^{(t)} + \eta \nabla_{\mathbf{S}} q_n(\widehat{\mathbf{S}}^{(t)}, \widehat{\mathbf{Z}}^{(t)})$ and $\widehat{\mathbf{S}}^{(t+1)}$ preserves the $s$ largest magnitudes in $\widehat{\mathbf{S}}^{(t+0.5)}$, it's easy to verify that

$$\widehat{\mathbf{S}}^{(t+1)} = \mathcal{HT}_s(\widehat{\mathbf{S}}^{(t+0.5)}) = \mathcal{HT}_s\big(\widehat{\mathbf{S}}^{(t)} + \eta\big[\nabla_{\mathbf{S}} q_n(\widehat{\mathbf{S}}^{(t)}, \widehat{\mathbf{Z}}^{(t)})\big]_{\bar{\mathcal{S}}}\big).$$

Thus by Lemma B.4 we have

$$\big\|\widehat{\mathbf{S}}^{(t+1)} - \mathbf{S}^*\big\|_F^2 \leq \bigg(1 + \frac{2\sqrt{s^*}}{\sqrt{s - s^*}}\bigg)\big\|\widehat{\mathbf{S}}^{(t)} - \eta\big[\nabla_{\mathbf{S}} q_n(\widehat{\mathbf{S}}^{(t)}, \widehat{\mathbf{Z}}^{(t)})\big]_{\bar{\mathcal{S}}} - \mathbf{S}^*\big\|_F^2$$

$$\leq 2\bigg(1 + \frac{2\sqrt{s^*}}{\sqrt{s - s^*}}\bigg)\big\|\widehat{\mathbf{S}}^{(t)} - \eta\big[\nabla_{\mathbf{S}} q(\widehat{\mathbf{S}}^{(t)}, \widehat{\mathbf{Z}}^{(t)})\big]_{\bar{\mathcal{S}}} - \mathbf{S}^*\big\|_F^2$$

$$+ 2\eta^2\bigg(1 + \frac{2\sqrt{s^*}}{\sqrt{s - s^*}}\bigg)\big\|\big[\nabla_{\mathbf{S}} q_n(\widehat{\mathbf{S}}^{(t)}, \widehat{\mathbf{Z}}^{(t)}) - \nabla_{\mathbf{S}} q(\widehat{\mathbf{S}}^{(t)}, \widehat{\mathbf{Z}}^{(t)})\big]_{\bar{\mathcal{S}}}\big\|_F^2. \quad \text{(B.8)}$$

Note that $|\bar{\mathcal{S}}| \leq s^* + 2s$ and by Lemma B.5, we have with probability at least $1 - C/d$ that

$$\big\|\big[\nabla_{\mathbf{S}} q_n(\widehat{\mathbf{S}}^{(t)}, \widehat{\mathbf{Z}}^{(t)}) - \nabla_{\mathbf{S}} q(\widehat{\mathbf{S}}^{(t)}, \widehat{\mathbf{Z}}^{(t)})\big]_{\bar{\mathcal{S}}}\big\|_F \leq \sqrt{s^* + 2s}\big\|\nabla_{\mathbf{S}} q(\widehat{\mathbf{S}}^{(t)}, \widehat{\mathbf{Z}}^{(t)}) - \nabla_{\mathbf{S}} q_n(\widehat{\mathbf{S}}^{(t)}, \widehat{\mathbf{Z}}^{(t)})\big\|_{\infty,\infty}$$

$$\leq \sqrt{s^* + 2s}\epsilon_1(n, \delta), \quad \text{(B.9)}$$

where $\epsilon_1(n, \delta) = C\sqrt{\log d/n}$. By definition we have $\bar{\mathcal{S}} = \mathcal{S}^* \cup \mathcal{S}^{(t)} \cup \mathcal{S}^{(t+1)}$, which yields

$$\big\|\widehat{\mathbf{S}}^{(t)} - \eta\big[\nabla_{\mathbf{S}} q(\widehat{\mathbf{S}}^{(t)}, \widehat{\mathbf{Z}}^{(t)})\big]_{\bar{\mathcal{S}}} - \mathbf{S}^*\big\|_F^2 \leq \big\|\widehat{\mathbf{S}}^{(t)} - \eta\nabla_{\mathbf{S}} q(\widehat{\mathbf{S}}^{(t)}, \widehat{\mathbf{Z}}^{(t)}) - \mathbf{S}^*\big\|_F^2$$

$$\leq \bigg(1 - \frac{2\eta\mu L}{L + \mu}\bigg)\big\|\widehat{\mathbf{S}}^{(t)} - \mathbf{S}^*\big\|_F^2 + \frac{25\eta^2\gamma_2^2\sigma_{\max}}{8}d^2(\widehat{\mathbf{Z}}^{(t)}, \mathbf{Z}^*), \tag{B.10}$$

where the second inequality is due to Lemma B.2. Here $L = 4\nu^2, \mu = 1/(4\nu^2)$ and $\gamma_2 = 8\nu^2$. Submitting (B.9) and (B.10) into (B.8), we obtain with probability at least $1 - C/d$ that

$$\big\|\widehat{\mathbf{S}}^{(t+1)} - \mathbf{S}^*\big\|_F^2 \leq 2\bigg(1 + \frac{2\sqrt{s^*}}{\sqrt{s - s^*}}\bigg)\bigg\{\bigg(1 - \frac{2\eta\mu L}{L + \mu}\bigg)\big\|\widehat{\mathbf{S}}^{(t)} - \mathbf{S}^*\big\|_F^2 + \frac{25\eta^2\gamma_2^2\sigma_{\max}}{8}d^2(\widehat{\mathbf{Z}}^{(t)}, \mathbf{Z}^*)$$

$$+ \eta^2(s^* + 2s)\epsilon_1^2(n, \delta)\bigg\}. \tag{B.11}$$

On the other hand, let $\mathbf{Z}^{(t+1)} = \widehat{\mathbf{Z}}^{(t)} - \eta'\nabla_{\mathbf{Z}} q(\widehat{\mathbf{S}}^{(t)}, \widehat{\mathbf{Z}}^{(t)})$. We have

$$d^2\big(\widehat{\mathbf{Z}}^{(t+1)}, \mathbf{Z}^*\big) = \min_{\widetilde{\mathbf{Z}} \in \mathcal{U}}\big\|\widehat{\mathbf{Z}}^{(t+1)} - \widetilde{\mathbf{Z}}\big\|_F^2 \leq 2\big\|\widehat{\mathbf{Z}}^{(t+1)} - \mathbf{Z}^{(t+1)}\big\|_F^2 + 2\min_{\widetilde{\mathbf{Z}} \in \mathcal{U}}\big\|\mathbf{Z}^{(t+1)} - \widetilde{\mathbf{Z}}\big\|_F^2$$

$$= 2\big\|\widehat{\mathbf{Z}}^{(t+1)} - \mathbf{Z}^{(t+1)}\big\|_F^2 + 2d^2\big(\mathbf{Z}^{(t+1)}, \mathbf{Z}^*\big). \tag{B.12}$$

By Lemma B.3 we have

$$d^2\big(\mathbf{Z}^{(t+1)}, \mathbf{Z}^*\big) \leq \bigg(1 - \frac{\eta'\sigma_{\min}\mu L}{2(L + \mu)}\bigg)d^2(\widehat{\mathbf{Z}}^{(t)}, \mathbf{Z}^*) + \frac{25\eta'^2\gamma_1^2\sigma_{\max}}{8}\big\|\widehat{\mathbf{S}}^{(t)} - \mathbf{S}^*\big\|_F^2, \tag{B.13}$$

where $L = 4\nu^2$, $\mu = 1/(4\nu^2)$, and $\gamma_1 = 8\nu^2$. By Lemma B.5, we have with probability at least $1 - C'/d$ that

$$\left\|\widehat{\mathbf{Z}}^{(t+1)} - \mathbf{Z}^{(t+1)}\right\|_F = \eta'\left\|\nabla_{\mathbf{Z}} q_n\left(\widehat{\mathbf{S}}^{(t)}, \widehat{\mathbf{Z}}^{(t)}\right) - \nabla_{\mathbf{Z}} q\left(\widehat{\mathbf{S}}^{(t)}, \widehat{\mathbf{Z}}^{(t)}\right)\right\|_F \leq \eta'\epsilon_2(n,\delta), \qquad \text{(B.14)}$$

where $\epsilon_2(n,\delta) = C'\nu\sqrt{\sigma_{\max}}\sqrt{rd/n}$. Substituting (B.12) with (B.13) and (B.14), we obtain

$$d^2\left(\widehat{\mathbf{Z}}^{(t+1)}, \mathbf{Z}^*\right) \leq 2\left(1 - \frac{\eta'\sigma_{\min}\mu L}{2(L+\mu)}\right)d^2\left(\widehat{\mathbf{Z}}^{(t)}, \mathbf{Z}^*\right) + \frac{25\eta'^2\gamma_1^2\sigma_{\max}}{4}\|\widehat{\mathbf{S}}^{(t)} - \mathbf{S}^*\|_F^2 + 2\eta'^2\epsilon_2^2(n,\delta)$$
$$\text{(B.15)}$$

holds with probability at least $1 - C'/d$. Combining (B.11) and (B.15), we then have

$$\max\left\{\|\widehat{\mathbf{S}}^{(t+1)} - \mathbf{S}^*\|_F^2, d^2\left(\widehat{\mathbf{Z}}^{(t+1)}, \mathbf{Z}^*\right)\right\}$$

$$\leq 2\left(1 + \frac{2\sqrt{s^*}}{\sqrt{s-s^*}}\right)\underbrace{\max\left\{1 - \frac{2\eta\mu L}{L+\mu} + \frac{25\eta^2\gamma_2^2\sigma_{\max}}{8}, 1 - \frac{\eta'\sigma_{\min}\mu L}{2(L+\mu)} + \frac{25\eta'^2\gamma_1^2\sigma_{\max}}{8}\right\}}_{\rho}$$

$$\cdot \max\left\{\|\widehat{\mathbf{S}}^{(t)} - \mathbf{S}^*\|_F^2, d^2\left(\widehat{\mathbf{Z}}^{(t)}, \mathbf{Z}^*\right)\right\}$$

$$+ \underbrace{\max\left\{2\left(1 + \frac{2\sqrt{s^*}}{\sqrt{s-s^*}}\right)\eta^2(s^*+2s)\epsilon_1^2(n,\delta), 2\eta'^2\epsilon_2^2(n,\delta)\right\}}_{\tau} \qquad \text{(B.16)}$$

holds with probability at least $1 - \max\{C, C'\}/d$. Recall that by Lemma B.2 and Lemma B.3 we have $L = 4\nu^2$, $\mu = 1/(4\nu^2)$, $\gamma_1 = 8\nu^2$ and $\gamma_2 = 8\nu^2$. And by Lemma B.5, we have $\epsilon_1(n,\delta) = 2\sqrt{\log d/n}$, $\epsilon_2(n,\delta) = 4\nu\sqrt{\sigma_{\max}}\sqrt{rd/n}$. Note that in Lemma B.2 and Lemma B.3, we require the step sizes satisfy $\eta \leq 1/[16(L+\mu)]$ and $\eta' \leq 1/[16(L+\mu)\sigma_{\max}]$. In order to ensure the convergence of our algorithm, we require that $\rho < 1$. Thus we choose $\eta = C_0/(\sigma_{\max}\nu^2)$ and $\eta' = C_0\sigma_{\min}/(\sigma_{\max}\nu^4)$, where $C_0 > 0$ is a sufficient small constant. Then we have

$$\rho = \max\left\{1 - \frac{\eta}{\nu^2}, 1 - \frac{\eta'\sigma_{\min}}{\nu^2}\right\}, \qquad \tau = \max\left\{\frac{48C_0^2}{\sigma_{\max}^2\nu^4}\frac{s^*\log d}{n}, \frac{32C_0^2\sigma_{\min}^2}{\sigma_{\max}\nu^6}\frac{rd}{n}\right\}. \qquad \text{(B.17)}$$

When we choose the thresholding parameter as $s \geq \left(4(1/(2\sqrt{\rho}) - 1)^2 + 1\right)s^*$, it's easy to derive $2\left(1 + 2\sqrt{s^*/(s-s^*)}\right) \leq 1/\sqrt{\rho}$. Then we have

$$\max\left\{\|\widehat{\mathbf{S}}^{(t+1)} - \mathbf{S}^*\|_F^2, d^2\left(\widehat{\mathbf{Z}}^{(t+1)}, \mathbf{Z}^*\right)\right\} \leq \sqrt{\rho}\max\left\{\|\widehat{\mathbf{S}}^{(t)} - \mathbf{S}^*\|_F^2, d^2\left(\widehat{\mathbf{Z}}^{(t)}, \mathbf{Z}^*\right)\right\} + \tau$$
$$\leq \sqrt{\rho}R^2 + (1 - \sqrt{\rho})R^2 = R^2,$$

where in the second inequality we use the fact that when the sample size $n$ is sufficient large, we are able to ensure $\tau \leq (1 - \sqrt{\rho})R^2$. Therefore, we have $\widehat{\mathbf{S}}^{(t+1)} \in \mathbb{B}_F(\mathbf{S}^*, R)$ and $\widehat{\mathbf{Z}}^{(t+1)} \in \mathbb{B}_d(\mathbf{Z}^*, R)$. By mathematical induction, we have $\widehat{\mathbf{S}}^{(t)} \in \mathbb{B}_F(\mathbf{S}^*, R)$ and $\widehat{\mathbf{Z}}^{(t)} \in \mathbb{B}_d(\mathbf{Z}^*, R)$, for any $t = 0, 1, \ldots$

Since (B.16) holds uniformly for all $t$, we further obtain with probability at least $1 - C_1/d$ that

$$\max\left\{\|\widehat{\mathbf{S}}^{(t+1)} - \mathbf{S}^*\|_F^2, d^2\left(\widehat{\mathbf{Z}}^{(t+1)}, \mathbf{Z}^*\right)\right\} \leq \sqrt{\rho}\max\left\{\|\widehat{\mathbf{S}}^{(t)} - \mathbf{S}^*\|_F^2, d^2\left(\widehat{\mathbf{Z}}^{(t)}, \mathbf{Z}^*\right)\right\} + \tau$$
$$\leq \frac{\tau}{1 - \sqrt{\rho}} + \sqrt{\rho^{t+1}} \cdot R,$$

where $\rho$ and $\tau$ are defined in (B.17) and $C_1 = \max\{C, C'\}$ is a positive constant, which completes the proof. $\qquad\square$

## C  Proof of Supporting Lemmas in Appendix B

In this section, we prove the lemmas used in the proof of main theorem. We first lay out some useful lemmas. The first lemma is about the strong convexity and smoothness.

**Lemma C.1.** The population loss function $p(\mathbf{S}, \mathbf{L}^*)$ is $\mu$-strongly convex and $L$-smooth with respect to $\mathbf{S}$, namely,

$$\mu\|\mathbf{S} - \mathbf{S}^*\|_F^2 \leq \langle \nabla_{\mathbf{S}} p(\mathbf{S}, \mathbf{L}^*) - \nabla_{\mathbf{S}} p(\mathbf{S}^*, \mathbf{L}^*), \mathbf{S} - \mathbf{S}^* \rangle \leq L\|\mathbf{S} - \mathbf{S}^*\|_F^2,$$

for all $\mathbf{S} \in \mathbb{B}_F(\mathbf{S}^*, R)$, where $\mu = 1/(4\nu^2)$ and $L = 4\nu^2$. Similarly, $p(\mathbf{S}^*, \mathbf{L})$ is $\mu$-strongly convex and $L$-smooth with respect to $\mathbf{L}$:

$$\mu\|\mathbf{L} - \mathbf{L}^*\|_F^2 \leq \langle \nabla_{\mathbf{L}} p(\mathbf{S}^*, \mathbf{L}) - \nabla_{\mathbf{L}} p(\mathbf{S}^*, \mathbf{L}^*), \mathbf{L} - \mathbf{L}^* \rangle \leq L\|\mathbf{L} - \mathbf{L}^*\|_F^2,$$

for $\mathbf{L} = \mathbf{Z}\mathbf{Z}^\top, \mathbf{L}^* = \mathbf{Z}^*\mathbf{Z}^{*\top}$ and $\mathbf{Z} \in \mathbb{B}_d(\mathbf{Z}^*, R)$. Here we use $\nabla_{\mathbf{L}} p(\mathbf{S}, \mathbf{L})$ to denote the gradient of the loss function with respect to $\mathbf{L}$.

In the following lemma, we show that the first-order stability, i.e., Condition 4.4 on the population loss function holds for $\mathbf{S}$ and $\mathbf{L}$.

**Lemma C.2.** For all $\mathbf{S} \in \mathbb{B}_F(\mathbf{S}^*, R)$ and $\mathbf{Z} \in \mathbb{B}_d(\mathbf{Z}^*, R)$, by definition we have $\mathbf{L} = \mathbf{Z}\mathbf{Z}^\top$ and $\mathbf{L}^* = \mathbf{Z}^*\mathbf{Z}^{*\top}$. We have the following properties for gradient with respect to $\mathbf{S}$ and $\mathbf{L}$

$$\|\nabla_{\mathbf{L}} p(\mathbf{S}, \mathbf{L}) - \nabla_{\mathbf{L}} p(\mathbf{S}^*, \mathbf{L})\|_F \leq \gamma_1 \|\mathbf{S} - \mathbf{S}^*\|_F,$$
$$\|\nabla_{\mathbf{S}} p(\mathbf{S}, \mathbf{L}) - \nabla_{\mathbf{S}} p(\mathbf{S}, \mathbf{L}^*)\|_F \leq \gamma_2 \|\mathbf{L} - \mathbf{L}^*\|_F,$$

where $\gamma_1 = \gamma_2 = 8\nu^2$.

## C.1 Proof of Lemma B.2

*Proof.* Since $\mathbf{S}^{(t+0.5)} = \widehat{\mathbf{S}}^{(t)} - \eta \nabla_{\mathbf{S}} q(\widehat{\mathbf{S}}^{(t)}, \widehat{\mathbf{Z}}^{(t)})$, we have

$$
\begin{aligned}
\|\mathbf{S}^{(t+0.5)} - \mathbf{S}^*\|_F^2 &= \|\widehat{\mathbf{S}}^{(t)} - \eta \nabla_{\mathbf{S}} q(\widehat{\mathbf{S}}^{(t)}, \widehat{\mathbf{Z}}^{(t)}) - \mathbf{S}^*\|_F^2 \\
&\leq \|\widehat{\mathbf{S}}^{(t)} - \mathbf{S}^*\|_F^2 - 2\eta \langle \nabla_{\mathbf{S}} q(\widehat{\mathbf{S}}^{(t)}, \widehat{\mathbf{Z}}^{(t)}), \widehat{\mathbf{S}}^{(t)} - \mathbf{S}^* \rangle + \eta^2 \|\nabla_{\mathbf{S}} q(\widehat{\mathbf{S}}^{(t)}, \widehat{\mathbf{Z}}^{(t)})\|_F^2 \\
&= \|\widehat{\mathbf{S}}^{(t)} - \mathbf{S}^*\|_F^2 - 2\eta \underbrace{\langle \nabla_{\mathbf{S}} q(\widehat{\mathbf{S}}^{(t)}, \widehat{\mathbf{Z}}^{(t)}) - \nabla_{\mathbf{S}} q(\mathbf{S}^*, \widehat{\mathbf{Z}}^{(t)}), \widehat{\mathbf{S}}^{(t)} - \mathbf{S}^* \rangle}_{I_1}
\end{aligned}
$$

$$- 2\eta \underbrace{\langle \nabla_{\mathbf{S}} q(\mathbf{S}^*, \widehat{\mathbf{Z}}^{(t)}), \widehat{\mathbf{S}}^{(t)} - \mathbf{S}^* \rangle}_{I_2} + \eta^2 \underbrace{\|\nabla_{\mathbf{S}} q(\widehat{\mathbf{S}}^{(t)}, \widehat{\mathbf{Z}}^{(t)})\|_F^2}_{I_3}. \tag{C.1}$$

Since by Lemma C.1 $p(\mathbf{S}, \mathbf{Z}^*\mathbf{Z}^{*\top})$ is $\mu$-strongly convex and $L$-smooth regarding with $\mathbf{S}$ around $\mathbf{S}^*$, and note that $p(\mathbf{S}, \mathbf{Z}^*\mathbf{Z}^{*\top}) = q(\mathbf{S}, \mathbf{Z}^*)$, we also have that $q(\mathbf{S}, \mathbf{Z}^*)$ is $\mu$-strongly convex and $L$-smooth regarding with $\mathbf{S}$ around $\mathbf{S}^*$. For term $I_1$, applying Lemma E.1 yields

$$
\begin{aligned}
I_1 &= \langle \nabla_{\mathbf{S}} q(\widehat{\mathbf{S}}^{(t)}, \widehat{\mathbf{Z}}^{(t)}) - \nabla_{\mathbf{S}} q(\mathbf{S}^*, \widehat{\mathbf{Z}}^{(t)}), \widehat{\mathbf{S}}^{(t)} - \mathbf{S}^* \rangle \\
&\geq \frac{\mu L}{L + \mu} \|\widehat{\mathbf{S}}^{(t)} - \mathbf{S}^*\|_F^2 + \frac{1}{L + \mu} \|\nabla_{\mathbf{S}} q(\widehat{\mathbf{S}}^{(t)}, \widehat{\mathbf{Z}}^{(t)}) - \nabla_{\mathbf{S}} q(\mathbf{S}^*, \widehat{\mathbf{Z}}^{(t)})\|_F^2. \tag{C.2}
\end{aligned}
$$

For term $I_2$ in (C.1), noting that $\nabla_{\mathbf{S}} q(\mathbf{S}^*, \widehat{\mathbf{Z}}^*) = 0$ and the fact that $\nabla_{\mathbf{S}} q(\mathbf{S}, \mathbf{Z}) = \nabla_{\mathbf{\Omega}} q(\mathbf{\Omega})$ where $\mathbf{\Omega} = \mathbf{S} + \mathbf{L}$ and $\mathbf{L} = \mathbf{Z}\mathbf{Z}^\top$, we have

$$I_2 = \langle \nabla_{\mathbf{S}} q(\mathbf{S}^*, \widehat{\mathbf{Z}}^{(t)}) - \nabla_{\mathbf{S}} q(\mathbf{S}^*, \mathbf{Z}^*), \widehat{\mathbf{S}}^{(t)} - \mathbf{S}^* \rangle = \langle \nabla_{\mathbf{\Omega}} q(\mathbf{S}^* + \widehat{\mathbf{L}}^{(t)}) - \nabla_{\mathbf{\Omega}} q(\mathbf{S}^* + \mathbf{L}^*), \widehat{\mathbf{S}}^{(t)} - \mathbf{S}^* \rangle.$$

Applying mean value theorem we further obtain

$$
\begin{aligned}
I_2 &= \text{vec}(\widehat{\mathbf{L}}^{(t)} - \mathbf{L}^*)^\top \nabla_{\mathbf{\Omega}}^2 q(\mathbf{S}^* + (1-t)\mathbf{L}^* + t\widehat{\mathbf{L}}^{(t)}) \text{vec}(\widehat{\mathbf{S}}^{(t)} - \mathbf{S}^*) \\
&\geq \lambda_{\min}(\nabla_{\mathbf{\Omega}}^2 q(\mathbf{\Omega}^* + t(\widehat{\mathbf{L}}^{(t)} - \mathbf{L}^*))) \|\widehat{\mathbf{L}}^{(t)} - \mathbf{L}^*\|_F \cdot \|\widehat{\mathbf{S}}^{(t)} - \mathbf{S}^*\|_F, \tag{C.3}
\end{aligned}
$$

for some $t \in (0, 1)$. Easy calculation and the properties of Kronecker product yield

$$
\begin{aligned}
\lambda_{\min}(\nabla_{\mathbf{\Omega}}^2 q(\mathbf{\Omega}^* + t(\widehat{\mathbf{L}}^{(t)} - \mathbf{L}^*))) &= \lambda_{\min}((\mathbf{\Omega}^* + t(\widehat{\mathbf{L}}^{(t)} - \mathbf{L}^*))^{-1} \otimes (\mathbf{\Omega}^* + t(\widehat{\mathbf{L}}^{(t)} - \mathbf{L}^*))^{-1}) \\
&= (\lambda_{\max}(\mathbf{\Omega}^* + t(\widehat{\mathbf{L}}^{(t)} - \mathbf{L}^*)))^{-2} \\
&\geq (\nu + t\|\widehat{\mathbf{L}}^{(t)} - \mathbf{L}^*\|_2)^{-2} \\
&\geq \frac{1}{4\nu^2}. \tag{C.4}
\end{aligned}
$$

Finally, we are going to bound term $I_3$ in (C.1). Specifically, we have

$$\|\nabla_{\mathbf{S}}q(\widehat{\mathbf{S}}^{(t)},\widehat{\mathbf{Z}}^{(t)})\|_F^2 \le 2\|\nabla_{\mathbf{S}}q(\widehat{\mathbf{S}}^{(t)},\widehat{\mathbf{Z}}^{(t)}) - \nabla_{\mathbf{S}}q(\mathbf{S}^*,\widehat{\mathbf{Z}}^{(t)})\|_F^2 + 2\|\nabla_{\mathbf{S}}q(\mathbf{S}^*,\widehat{\mathbf{Z}}^{(t)}) - \nabla_{\mathbf{S}}q(\mathbf{S}^*,\mathbf{Z}^*)\|_F^2$$

$$= 2\|\nabla_{\mathbf{S}}q(\widehat{\mathbf{S}}^{(t)},\widehat{\mathbf{Z}}^{(t)}) - \nabla_{\mathbf{S}}q(\mathbf{S}^*,\widehat{\mathbf{Z}}^{(t)})\|_F^2 + 2\|\nabla_{\mathbf{S}}p(\mathbf{S}^*,\widehat{\mathbf{L}}^{(t)}) - \nabla_{\mathbf{S}}p(\mathbf{S}^*,\mathbf{L}^*)\|_F^2$$

$$\le 2\|\nabla_{\mathbf{S}}q(\widehat{\mathbf{S}}^{(t)},\widehat{\mathbf{Z}}^{(t)}) - \nabla_{\mathbf{S}}q(\mathbf{S}^*,\widehat{\mathbf{Z}}^{(t)})\|_F^2 + 2\gamma_2^2\|\widehat{\mathbf{L}}^{(t)} - \mathbf{L}^*\|_F^2, \qquad (C.5)$$

where the first inequality is due to $(a+b)^2 \le 2a^2 + 2b^2$, the equality is due $q(\mathbf{S},\mathbf{Z}) = p(\mathbf{S},\mathbf{Z}\mathbf{Z}^\top) = p(\mathbf{S},\mathbf{L})$, and the last inequality is by the first-order stability property, i.e., Lemma C.2, where $\gamma_2 = 8\nu^2$. Submitting (C.2), (C.3), (C.4) and (C.5) into (C.1) yields

$$\|\mathbf{S}^{(t+0.5)} - \mathbf{S}^*\|_F^2 \le \left(1 - \frac{2\eta\mu L}{L+\mu}\right)\|\widehat{\mathbf{S}}^{(t)} - \mathbf{S}^*\|_F^2 + 2\eta\left(\eta - \frac{1}{L+\mu}\right)\|\nabla_{\mathbf{S}}q(\widehat{\mathbf{S}}^{(t)},\widehat{\mathbf{Z}}^{(t)}) - \nabla_{\mathbf{S}}q(\mathbf{S}^*,\widehat{\mathbf{Z}}^{(t)})\|_F^2$$

$$- \frac{\eta}{2\nu^2}\|\widehat{\mathbf{L}}^{(t)} - \mathbf{L}^*\|_F \cdot \|\widehat{\mathbf{S}}^{(t)} - \mathbf{S}^*\|_F + 2\eta^2\gamma_2^2\|\widehat{\mathbf{L}}^{(t)} - \mathbf{L}^*\|_F^2. \qquad (C.6)$$

Noting that $\|\widehat{\mathbf{L}}^{(t)} - \mathbf{L}^*\|_F \le (R + \sqrt{\sigma_{\max}})d(\widehat{\mathbf{Z}}^{(t)},\mathbf{Z}^*) \le 5/4\sqrt{\sigma_{\max}}d(\widehat{\mathbf{Z}}^{(t)},\mathbf{Z}^*)$, by setting $\eta \le 1/(L+\mu)$ we have

$$\|\mathbf{S}^{(t+0.5)} - \mathbf{S}^*\|_F^2 \le \left(1 - \frac{2\eta\mu L}{L+\mu}\right)\|\widehat{\mathbf{S}}^{(t)} - \mathbf{S}^*\|_F^2 + \frac{25\eta^2\gamma_2^2\sigma_{\max}}{8}d^2(\widehat{\mathbf{Z}}^{(t)},\mathbf{Z}^*). \qquad (C.7)$$

$\square$

## C.2 Proof of Lemma B.3

*Proof.* Based on the definition in (4.3) we denote

$$\bar{\mathbf{Z}}^{(t)} = \underset{\widetilde{\mathbf{Z}}\in\mathcal{U}}{\operatorname{argmin}} \|\widehat{\mathbf{Z}}^{(t)} - \widetilde{\mathbf{Z}}\|_F,$$

which implies $d(\widehat{\mathbf{Z}}^{(t)},\mathbf{Z}^*) = \min_{\widetilde{\mathbf{Z}}\in\mathcal{U}}\|\widehat{\mathbf{Z}}^{(t)} - \widetilde{\mathbf{Z}}\|_F = \|\widehat{\mathbf{Z}}^{(t)} - \bar{\mathbf{Z}}^{(t)}\|_F$. Thus by defining $\mathbf{Z}^{(t+1)} = \widehat{\mathbf{Z}}^{(t)} - \eta'\nabla_{\mathbf{Z}}q(\widehat{\mathbf{S}}^{(t)},\widehat{\mathbf{Z}}^{(t)})$ as the population version of $\widehat{\mathbf{Z}}^{(t+1)}$, we have

$$d(\mathbf{Z}^{(t+1)},\mathbf{Z}^*) = \min_{\widetilde{\mathbf{Z}}\in\mathcal{U}}\|\mathbf{Z}^{(t+1)} - \widetilde{\mathbf{Z}}\|_F \le \|\mathbf{Z}^{(t+1)} - \bar{\mathbf{Z}}^{(t)}\|_F,$$

it follows that

$$d^2(\mathbf{Z}^{(t+1)},\mathbf{Z}^*) \le \|\widehat{\mathbf{Z}}^{(t)} - \eta'\nabla_{\mathbf{Z}}q(\widehat{\mathbf{S}}^{(t)},\widehat{\mathbf{Z}}^{(t)}) - \bar{\mathbf{Z}}^{(t)}\|_F^2$$

$$= d^2(\widehat{\mathbf{Z}}^{(t)},\mathbf{Z}^*) - 2\eta'\underbrace{\langle\nabla_{\mathbf{Z}}q(\widehat{\mathbf{S}}^{(t)},\widehat{\mathbf{Z}}^{(t)}),\widehat{\mathbf{Z}}^{(t)} - \bar{\mathbf{Z}}^{(t)}\rangle}_{I_1} + \eta'^2\underbrace{\|\nabla_{\mathbf{Z}}q(\widehat{\mathbf{S}}^{(t)},\widehat{\mathbf{Z}}^{(t)})\|_F^2}_{I_2}.$$

$$(C.8)$$

For term $I_1$ in (C.8), note that we have $\nabla_{\mathbf{Z}}q(\widehat{\mathbf{S}}^{(t)},\widehat{\mathbf{Z}}^{(t)}) = [\nabla_{\mathbf{L}}p(\widehat{\mathbf{S}}^{(t)},\widehat{\mathbf{L}}^{(t)})]\widehat{\mathbf{Z}}^{(t)}, \widehat{\mathbf{L}}^{(t)} = \widehat{\mathbf{Z}}^{(t)}[\widehat{\mathbf{Z}}^{(t)}]^\top$ and $\mathbf{L}^* = \bar{\mathbf{Z}}^{(t)}[\bar{\mathbf{Z}}^{(t)}]^\top$. It follows that

$$\langle\nabla_{\mathbf{Z}}q(\widehat{\mathbf{S}}^{(t)},\widehat{\mathbf{Z}}^{(t)}),\widehat{\mathbf{Z}}^{(t)} - \bar{\mathbf{Z}}^{(t)}\rangle = \langle\nabla_{\mathbf{L}}p(\widehat{\mathbf{S}}^{(t)},\widehat{\mathbf{L}}^{(t)}),\widehat{\mathbf{Z}}^{(t)}[\widehat{\mathbf{Z}}^{(t)} - \bar{\mathbf{Z}}^{(t)}]^\top\rangle$$

$$= \langle\nabla_{\mathbf{L}}p(\widehat{\mathbf{S}}^{(t)},\mathbf{L}^*),\widehat{\mathbf{Z}}^{(t)}[\widehat{\mathbf{Z}}^{(t)} - \bar{\mathbf{Z}}^{(t)}]^\top\rangle + \langle\nabla_{\mathbf{L}}p(\widehat{\mathbf{S}}^{(t)},\widehat{\mathbf{L}}^{(t)}) - \nabla_{\mathbf{L}}p(\widehat{\mathbf{S}}^{(t)},\mathbf{L}^*),\widehat{\mathbf{Z}}^{(t)}[\widehat{\mathbf{Z}}^{(t)} - \bar{\mathbf{Z}}^{(t)}]^\top\rangle$$

$$= \underbrace{\frac{1}{2}\langle\nabla_{\mathbf{L}}p(\widehat{\mathbf{S}}^{(t)},\mathbf{L}^*),\widehat{\mathbf{L}}^{(t)} - \mathbf{L}^* + [\widehat{\mathbf{Z}}^{(t)} - \bar{\mathbf{Z}}^{(t)}][\widehat{\mathbf{Z}}^{(t)} - \bar{\mathbf{Z}}^{(t)}]^\top\rangle}_{I_{11}} + \underbrace{\frac{1}{2}\langle\nabla_{\mathbf{L}}p(\widehat{\mathbf{S}}^{(t)},\widehat{\mathbf{L}}^{(t)}) - \nabla_{\mathbf{L}}p(\widehat{\mathbf{S}}^{(t)},\mathbf{L}^*),\widehat{\mathbf{L}}^{(t)} - \mathbf{L}^*\rangle}_{I_{12}}$$

$$+ \underbrace{\frac{1}{2}\langle\nabla_{\mathbf{L}}p(\widehat{\mathbf{S}}^{(t)},\widehat{\mathbf{L}}^{(t)}) - \nabla_{\mathbf{L}}p(\widehat{\mathbf{S}}^{(t)},\mathbf{L}^*),[\widehat{\mathbf{Z}}^{(t)} - \bar{\mathbf{Z}}^{(t)}][\widehat{\mathbf{Z}}^{(t)} - \bar{\mathbf{Z}}^{(t)}]^\top\rangle}_{I_{13}}. \qquad (C.9)$$

We first bound term $I_{11}$ in (C.9). Noting that $\nabla_{\mathbf{L}}p(\mathbf{S},\mathbf{L}) = \nabla_{\mathbf{\Omega}}p(\mathbf{\Omega})$, where $\mathbf{\Omega} = \mathbf{S} + \mathbf{L}$ and $\mathbf{L} = \mathbf{Z}\mathbf{Z}^\top$, we obtain

$$I_{11} = \frac{1}{2}\langle\nabla_{\mathbf{L}}p(\widehat{\mathbf{S}}^{(t)},\mathbf{L}^*),\widehat{\mathbf{L}}^{(t)} - \mathbf{L}^* + [\widehat{\mathbf{Z}}^{(t)} - \bar{\mathbf{Z}}^{(t)}][\widehat{\mathbf{Z}}^{(t)} - \bar{\mathbf{Z}}^{(t)}]^\top\rangle$$

$$= \frac{1}{2}\langle\nabla_{\mathbf{\Omega}}p(\widehat{\mathbf{S}}^{(t)} + \mathbf{L}^*) - \nabla_{\mathbf{\Omega}}p(\mathbf{\Omega}^*),\widehat{\mathbf{L}}^{(t)} - \mathbf{L}^* + [\widehat{\mathbf{Z}}^{(t)} - \bar{\mathbf{Z}}^{(t)}][\widehat{\mathbf{Z}}^{(t)} - \bar{\mathbf{Z}}^{(t)}]^\top\rangle,$$

where we used the fact that $\nabla_{\boldsymbol{\Omega}} p(\boldsymbol{\Omega}^*) = \mathbf{0}$. Applying mean value theorem yields

$$I_{11} = 1/2 \text{vec} \big( \widehat{\mathbf{S}}^{(t)} - \mathbf{S}^* \big)^\top \nabla_{\boldsymbol{\Omega}}^2 p(\mathbf{L}^* + (1-t)\mathbf{S}^* + t\widehat{\mathbf{S}}^{(t)}) \text{vec} \big( \widehat{\mathbf{L}}^{(t)} - \mathbf{L}^* + \big[ \widehat{\mathbf{Z}}^{(t)} - \bar{\mathbf{Z}}^{(t)} \big] \big[ \widehat{\mathbf{Z}}^{(t)} - \bar{\mathbf{Z}}^{(t)} \big]^\top \big)$$
$$\geq 1/2 \lambda_{\min} \big( \nabla_{\boldsymbol{\Omega}}^2 p(\boldsymbol{\Omega}^* + t(\widehat{\mathbf{S}}^{(t)} - \mathbf{S}^*)) \big) \big\| \widehat{\mathbf{S}}^{(t)} - \mathbf{S}^* \big\|_F \cdot \big\| \widehat{\mathbf{L}}^{(t)} - \mathbf{L}^* + \big[ \widehat{\mathbf{Z}}^{(t)} - \bar{\mathbf{Z}}^{(t)} \big] \big[ \widehat{\mathbf{Z}}^{(t)} - \bar{\mathbf{Z}}^{(t)} \big]^\top \big\|_F, \tag{C.10}$$

for some $t \in (0,1)$. Simple calculation yields

$$\lambda_{\min} \big( \nabla_{\boldsymbol{\Omega}}^2 p(\boldsymbol{\Omega}^* + t(\widehat{\mathbf{S}}^{(t)} - \mathbf{S}^*)) \big) = \lambda_{\min} \big( (\boldsymbol{\Omega}^* + t(\widehat{\mathbf{S}}^{(t)} - \mathbf{S}^*))^{-1} \otimes (\boldsymbol{\Omega}^* + t(\widehat{\mathbf{S}}^{(t)} - \mathbf{S}^*)^{-1}) \big)$$
$$= \| \boldsymbol{\Omega}^* + t(\widehat{\mathbf{S}}^{(t)} - \mathbf{S}^*) \|_2^{-2}$$
$$\geq \frac{1}{4\nu^2}. \tag{C.11}$$

Thus, combining (C.10) and (C.11) we obtain

$$I_{11} \geq \frac{1}{8\nu^2} \big\| \widehat{\mathbf{S}}^{(t)} - \mathbf{S}^* \big\|_F \cdot \big\| \widehat{\mathbf{L}}^{(t)} - \mathbf{L}^* + \big[ \widehat{\mathbf{Z}}^{(t)} - \bar{\mathbf{Z}}^{(t)} \big] \big[ \widehat{\mathbf{Z}}^{(t)} - \bar{\mathbf{Z}}^{(t)} \big]^\top \big\|_F. \tag{C.12}$$

Next, since by Lemma C.1 $p(\widehat{\mathbf{S}}^{(t)}, \mathbf{L})$ is $\mu$-strongly convex and $L$-smooth with respect to $\mathbf{L}$ with $\mu = 1/(4\nu^2)$ and $L = 4\nu^2$, by Lemma E.1 we further obtain

$$I_{12} \geq \frac{\mu L}{2(\mu + L)} \big\| \widehat{\mathbf{L}}^{(t)} - \mathbf{L}^* \big\|_F^2 + \frac{1}{2(\mu + L)} \big\| \nabla_{\mathbf{L}} p(\widehat{\mathbf{S}}^{(t)}, \widehat{\mathbf{L}}^{(t)}) - \nabla_{\mathbf{L}} p(\widehat{\mathbf{S}}^{(t)}, \mathbf{L}^*) \big\|_F^2. \tag{C.13}$$

For term $I_{13}$ in (C.9), we have

$$I_{13} \geq -\frac{1}{2} \big\| \nabla_{\mathbf{L}} p(\widehat{\mathbf{S}}^{(t)}, \widehat{\mathbf{L}}^{(t)}) - \nabla_{\mathbf{L}} p(\widehat{\mathbf{S}}^{(t)}, \mathbf{L}^*) \big\|_F \cdot \big\| \bar{\mathbf{Z}}^{(t)} - \widehat{\mathbf{Z}}^{(t)} \big\|_F^2$$
$$\geq -\frac{1}{4c} \big\| \nabla_{\mathbf{L}} p(\widehat{\mathbf{S}}^{(t)}, \widehat{\mathbf{L}}^{(t)}) - \nabla_{\mathbf{L}} p(\widehat{\mathbf{S}}^{(t)}, \mathbf{L}^*) \big\|_F^2 - \frac{c}{4} \big\| \bar{\mathbf{Z}}^{(t)} - \widehat{\mathbf{Z}}^{(t)} \big\|_F^4, \tag{C.14}$$

where in the second inequality we used the inequality $2ab \leq a^2/c + cb^2$ for any $c > 0$.

Now we turn to term $I_2$ in (C.8). Recall that $\nabla_{\mathbf{Z}} q(\mathbf{S}, \mathbf{Z}) = [\nabla_{\mathbf{L}} p(\mathbf{S}, \mathbf{L})]\mathbf{Z}$. We have

$$\| \nabla_{\mathbf{Z}} q(\widehat{\mathbf{S}}^{(t)}, \widehat{\mathbf{Z}}^{(t)}) \|_F^2 \leq 2\|[\nabla_{\mathbf{L}} p(\widehat{\mathbf{S}}^{(t)}, \widehat{\mathbf{L}}^{(t)}) - \nabla_{\mathbf{L}} p(\widehat{\mathbf{S}}^{(t)}, \mathbf{L}^*)]\widehat{\mathbf{Z}}^{(t)}\|_F^2 + 2\|[\nabla_{\mathbf{L}} p(\widehat{\mathbf{S}}^{(t)}, \mathbf{L}^*) - \nabla_{\mathbf{L}} p(\mathbf{S}^*, \mathbf{L}^*)]\widehat{\mathbf{Z}}^{(t)}\|_F^2$$
$$\leq 2\| \nabla_{\mathbf{L}} p(\widehat{\mathbf{S}}^{(t)}, \widehat{\mathbf{L}}^{(t)}) - \nabla_{\mathbf{L}} p(\widehat{\mathbf{S}}^{(t)}, \mathbf{L}^*) \|_F^2 \cdot \| \widehat{\mathbf{Z}}^{(t)} \|_2^2 + 2\gamma_1^2 \| \widehat{\mathbf{S}}^{(t)} - \mathbf{S}^* \|_F^2 \cdot \| \widehat{\mathbf{Z}}^{(t)} \|_2^2$$
$$\leq \frac{25\sigma_{\max}}{8} \| \nabla_{\mathbf{L}} p(\widehat{\mathbf{S}}^{(t)}, \widehat{\mathbf{L}}^{(t)}) - \nabla_{\mathbf{L}} p(\widehat{\mathbf{S}}^{(t)}, \mathbf{L}^*) \|_F^2 + \frac{25\gamma_1^2 \sigma_{\max}}{8} \| \widehat{\mathbf{S}}^{(t)} - \mathbf{S}^* \|_F^2, \tag{C.15}$$

where the second inequality is due to Lemma C.2 with $\gamma_1 = 8\nu^2$, and the last inequlity is due to $\|\widehat{\mathbf{Z}}^{(t)}\|_2 \leq \|\mathbf{Z}^*\|_2 + d(\widehat{\mathbf{Z}}^{(t)}, \mathbf{Z}^*) \leq R + \sqrt{\sigma_{\max}} \leq 5/4\sqrt{\sigma_{\max}}$.

Thus submitting (C.12), (C.13), (C.14) and (C.15) into (C.8) yields

$$d^2(\mathbf{Z}^{(t+1)}, \mathbf{Z}^*) \leq \left( 1 - \frac{2\eta'(\sqrt{2}-1)\sigma_{\min}\mu L}{L+\mu} \right) d^2(\widehat{\mathbf{Z}}^{(t)}, \mathbf{Z}^*) + \frac{c\eta'}{2} \big\| \bar{\mathbf{Z}}^{(t)} - \widehat{\mathbf{Z}}^{(t)} \big\|_F^4 + \frac{25\eta'^2\gamma_1^2\sigma_{\max}}{8} \| \widehat{\mathbf{S}}^{(t)} - \mathbf{S}^* \|_F^2$$
$$+ \left( \frac{25\eta'^2\sigma_{\max}}{8} + \frac{\eta'}{2c} - \frac{\eta'}{L+\mu} \right) \big\| \nabla_{\mathbf{L}} p(\widehat{\mathbf{S}}^{(t)}, \widehat{\mathbf{L}}^{(t)}) - \nabla_{\mathbf{L}} p(\widehat{\mathbf{S}}^{(t)}, \mathbf{L}^*) \big\|_F^2$$
$$- \frac{\eta'}{4\nu^2} \big\| \widehat{\mathbf{S}}^{(t)} - \mathbf{S}^* \big\|_F \cdot \big\| \widehat{\mathbf{L}}^{(t)} - \mathbf{L}^* + \big[ \widehat{\mathbf{Z}}^{(t)} - \bar{\mathbf{Z}}^{(t)} \big] \big[ \widehat{\mathbf{Z}}^{(t)} - \bar{\mathbf{Z}}^{(t)} \big]^\top \big\|_F$$
$$\leq \left( 1 - \frac{2\eta'(\sqrt{2}-1)\sigma_{\min}\mu L}{L+\mu} + \frac{\eta' R^2(L+\mu)}{2} \right) d^2(\widehat{\mathbf{Z}}^{(t)}, \mathbf{Z}^*) + \frac{25\eta'^2\gamma_1^2\sigma_{\max}}{8} \| \widehat{\mathbf{S}}^{(t)} - \mathbf{S}^* \|_F^2, \tag{C.16}$$

where in the first inequality we used the conclusion in Lemma E.4 and that $\sigma_{\min}(\mathbf{Z}^*) = \sqrt{\sigma_{\min}}$; in the second inequality we chose $c = L + \mu$ and used the condition that $\eta' \leq 4/[25(L+\mu)\sigma_{\max}]$. By our condition that $R \leq \sqrt{\sigma_{\min}}/(6.5\nu^2)$, we get $R^2 \leq 3\sigma_{\min}/(125\nu^4) \leq (4\sqrt{2}-5)\sigma_{\min}\mu L/(L+\mu)^2$, which immediately implies

$$d^2(\mathbf{Z}^{(t+1)}, \mathbf{Z}^*) \leq \left( 1 - \frac{\eta'\sigma_{\min}\mu L}{2(L+\mu)} \right) d^2(\widehat{\mathbf{Z}}^{(t)}, \mathbf{Z}^*) + \frac{25\eta'^2\gamma_1^2\sigma_{\max}}{8} \| \widehat{\mathbf{S}}^{(t)} - \mathbf{S}^* \|_F^2, \tag{C.17}$$

which completes the proof. $\qquad\square$

## C.3 Proof of Lemma B.5

Now we are going to prove the lemma of statistical errors.

*Proof.* This lemma has two parts: one is the statistical error for the derivatives of loss functions with respect to $\mathbf{S}$, and the other one with respect to $\mathbf{Z}$. We first deal with $\mathbf{S}$.

**Part 1:** Taking derivative of $q(\mathbf{S}, \mathbf{Z})$ with respect to $\mathbf{S}$ while fixing $\mathbf{Z}$, we have

$$\nabla_{\mathbf{S}} q(\mathbf{S}, \mathbf{Z}) = \mathbf{\Sigma}^* - (\mathbf{S} + \mathbf{Z}\mathbf{Z}^\top)^{-1}.$$

Take derivative of $q_n(\mathbf{S}, \mathbf{Z})$ with respect to $\mathbf{S}$ while fixing $\mathbf{Z}$, we have

$$\nabla_{\mathbf{S}} q_n(\mathbf{S}, \mathbf{Z}) = \widehat{\mathbf{\Sigma}} - (\mathbf{S} + \mathbf{Z}\mathbf{Z}^\top)^{-1} = \frac{1}{n} \sum_{i=1}^n \mathbf{X}_i \mathbf{X}_i^\top - (\mathbf{S} + \mathbf{Z}\mathbf{Z}^\top)^{-1}.$$

Thus by Lemma E.2, we obtain

$$\left\| \nabla_{\mathbf{S}} q_n(\mathbf{S}, \mathbf{Z}) - \nabla_{\mathbf{S}} q(\mathbf{S}, \mathbf{Z}) \right\|_{\infty, \infty} = \left\| \frac{1}{n} \sum_{i=1}^n \mathbf{X}_i \mathbf{X}_i^\top - \mathbf{\Sigma}^* \right\|_{\infty, \infty} \leq 2\sqrt{\frac{\log d}{n}} \qquad (\text{C.18})$$

holds with probability at least $1 - C/d$.

**Part 2:** Taking derivative of $q(\mathbf{S}, \mathbf{Z})$ with respect to $\mathbf{Z}$ while fixing $\mathbf{S}$, we have

$$\nabla_{\mathbf{Z}} q(\mathbf{S}, \mathbf{Z}) = 2\mathbf{\Sigma}^* \mathbf{Z} - 2(\mathbf{S} + \mathbf{Z}\mathbf{Z}^\top)^{-1} \mathbf{Z}.$$

Taking derivative of $q_n(\mathbf{S}, \mathbf{Z})$ with respect to $\mathbf{Z}$ while fixing $\mathbf{S}$, we have

$$\nabla_{\mathbf{Z}} q_n(\mathbf{S}, \mathbf{Z}) = 2\widehat{\mathbf{\Sigma}} \mathbf{Z} - 2(\mathbf{S} + \mathbf{Z}\mathbf{Z}^\top)^{-1} \mathbf{Z}.$$

Then by transformation of norm, we have

$$\|\nabla_{\mathbf{Z}} q_n(\mathbf{S}, \mathbf{Z}) - \nabla_{\mathbf{Z}} q(\mathbf{S}, \mathbf{Z})\|_F = 2\left\| (\widehat{\mathbf{\Sigma}} - \mathbf{\Sigma}^*) \mathbf{Z} \right\|_F \leq 2 \left\| \frac{1}{n} \sum_{i=1}^n \mathbf{X}_i \mathbf{X}_i^\top - \mathbf{\Sigma}^* \right\|_2 \cdot \|\mathbf{Z}\|_F.$$

Since $\|\mathbf{Z}\|_F \leq \|\mathbf{Z} - \bar{\mathbf{Z}}\|_F + \|\bar{\mathbf{Z}}\|_F$ and $\|\mathbf{Z} - \bar{\mathbf{Z}}\|_F = d(\mathbf{Z}, \mathbf{Z}^*) \leq R$, $\|\bar{\mathbf{Z}}\|_2 = \|\mathbf{Z}^*\|_2 \leq \sqrt{\sigma_{\max}}$, we have $\|\mathbf{Z}\|_F \leq R + \sqrt{r\sigma_{\max}}$. Lemma E.3 shows that we have

$$\left\| \frac{1}{n} \sum_{i=1}^n \mathbf{X}_i \mathbf{X}_i^\top - \mathbf{\Sigma}^* \right\|_2 \leq 2\nu \sqrt{\frac{d}{n}}$$

with probability at least $1 - C'/d$. It immediately follows that

$$\|\nabla_{\mathbf{Z}} q_n(\mathbf{S}, \mathbf{Z}) - \nabla_{\mathbf{Z}} q(\mathbf{S}, \mathbf{Z})\|_F \leq 4\nu\sqrt{\sigma_{\max}}\sqrt{\frac{rd}{n}}$$

holds with probability at least $1 - C'/d$. $\qquad \square$

## C.4 Proof of Lemma B.1

*Proof.* By definition we have $\|\mathbf{A}\|_2 = \sup_{\|\mathbf{x}\|_2 = 1} \mathbf{x}^\top \mathbf{A}\mathbf{x}$. Note that

$$\mathbf{x}^\top \mathbf{A}\mathbf{x} = \langle \mathbf{x}, \mathbf{A}\mathbf{x} \rangle = \langle \mathbf{x}\mathbf{x}^\top, \mathbf{A} \rangle \leq \|\mathbf{x}\mathbf{x}^\top\|_F \cdot \|\mathbf{A}\|_F \leq \sqrt{s_0}\|\mathbf{A}\|_{\infty, \infty},$$

where in the last inequality we use the fact that $\|\mathbf{x}\mathbf{x}^\top\|_F = 1$. $\qquad \square$

# D Proof of Additional Lemmas in Appendix C

## D.1 Proof of Lemma C.1

*Proof.* We first show the strong convexity and smoothness with respect to $\mathbf{S}$. Taking derivative of $p(\mathbf{S}, \mathbf{L}^*)$ with respect to $\mathbf{S}$ while fixing $\mathbf{L}^*$ and denoting the gradient as $\nabla_{\mathbf{S}} p(\mathbf{S}, \mathbf{L}^*)$, we have

$$\nabla_{\mathbf{S}} p(\mathbf{S}, \mathbf{L}^*) = \mathbf{\Sigma}^* - (\mathbf{S} + \mathbf{L}^*)^{-1}.$$

Further, taking the second order derivative with respect to $\mathbf{S}$, we get

$$\nabla_{\mathbf{S}}^2 p(\mathbf{S}, \mathbf{L}^*) = (\mathbf{S} + \mathbf{L}^*)^{-1} \otimes (\mathbf{S} + \mathbf{L}^*)^{-1}. \tag{D.1}$$

For any $\mathbf{S} \in \mathbb{B}_F(\mathbf{S}^*, R)$, we define

$$\mathcal{E}(\mathbf{S}) = \langle \nabla_{\mathbf{S}} p(\mathbf{S}, \mathbf{L}^*) - \nabla_{\mathbf{S}} p(\mathbf{S}^*, \mathbf{L}^*), \mathbf{S} - \mathbf{S}^* \rangle. \tag{D.2}$$

Applying mean value theorem to (D.2), we obtain

$$\mathcal{E}(\mathbf{S}) \geq \lambda_{\min}(\nabla_{\mathbf{S}}^2 p(\mathbf{S}^* + \theta(\mathbf{S} - \mathbf{S}^*), \mathbf{L}^*)) \|\mathbf{S} - \mathbf{S}^*\|_F^2 = \lambda_{\max}(\mathbf{S}^* + \theta(\mathbf{S} - \mathbf{S}^*) + \mathbf{L}^*)^{-2} \|\mathbf{S} - \mathbf{S}^*\|_F^2, \tag{D.3}$$

for some $\theta \in [0, 1]$, where in the last equality we use the property of Kronecker product. By triangle inequality we have

$$\lambda_{\max}(\mathbf{S}^* + \theta(\mathbf{S} - \mathbf{S}^*) + \mathbf{L}^*) \leq \|\mathbf{S}^* + \mathbf{L}^*\|_2 + \theta \|\mathbf{S} - \mathbf{S}^*\|_2 \leq \nu + R \leq 2\nu, \tag{D.4}$$

where the last inequality is because we have $R \leq 1/\nu \leq \nu$ by definition. Combining (D.3) and (D.4) yields

$$\mathcal{E}(\mathbf{S}) \geq \frac{1}{4\nu^2} \|\mathbf{S} - \mathbf{S}^*\|_F^2,$$

which immediately implies that $q(\mathbf{S}, \mathbf{L}^*)$ is $\mu$-strongly convex with respect to $\mathbf{S}$, where $\mu = 1/(4\nu^2)$.

Note that for $\mathcal{E}(\mathbf{S})$ defined in (D.2), we also have

$$\mathcal{E}(\mathbf{S}) \leq \lambda_{\max}(\nabla_{\mathbf{S}}^2 p(\mathbf{S}^* + \theta(\mathbf{S} - \mathbf{S}^*), \mathbf{L}^*)) \|\mathbf{S} - \mathbf{S}^*\|_F^2 = \lambda_{\min}(\mathbf{S}^* + \theta(\mathbf{S} - \mathbf{S}^*) + \mathbf{L}^*)^{-2} \|\mathbf{S} - \mathbf{S}^*\|_F^2, \tag{D.5}$$

For any $\mathbf{x} \in \mathbb{R}^d$ such that $\|\mathbf{x}\|_2 = 1$, we have

$$\mathbf{x}^\top (\mathbf{S}^* + \theta(\mathbf{S} - \mathbf{S}^*) + \mathbf{L}^*)\mathbf{x} = (1 - \theta)\mathbf{x}^\top (\mathbf{S} - \mathbf{S}^*)\mathbf{x} + \mathbf{x}(\mathbf{S}^* + \mathbf{L}^*)\mathbf{x} \geq -(1 - \theta)|\mathbf{x}^\top (\mathbf{S} - \mathbf{S}^*)\mathbf{x}| + \mathbf{x}(\mathbf{S}^* + \mathbf{L}^*)\mathbf{x},$$

where the last inequality is due to $0 \leq \theta \leq 1$. Taking minimization over $\mathbf{x}$ on both side of the inequality above, we have

$$\begin{aligned} \lambda_{\min}(\mathbf{S}^* + \theta(\mathbf{S} - \mathbf{S}^*) + \mathbf{L}^*) &= \min_{\|\mathbf{x}\|_2=1} \mathbf{x}^\top (\mathbf{S}^* + \theta(\mathbf{S} - \mathbf{S}^*) + \mathbf{L}^*)\mathbf{x} \\ &\geq (1 - \theta) \min_{\|\mathbf{x}\|_2=1} \left\{ -|\mathbf{x}^\top (\mathbf{S} - \mathbf{S}^*)\mathbf{x}| \right\} + \min_{\|\mathbf{x}\|_2=1} \mathbf{x}(\mathbf{S}^* + \mathbf{L}^*)\mathbf{x} \\ &\geq \frac{1}{\nu} - R \geq \frac{1}{2\nu}, \end{aligned}$$

where in the last inequality we use the fact $\mathbf{S} \in \mathbb{B}_F(\mathbf{S}^*, R)$ and $R \leq 1/(2\nu)$. Then it follows that

$$\mathcal{E}(\mathbf{S}) \leq 4\nu^2 \|\mathbf{S}' - \mathbf{S}\|_F^2,$$

which immediately implies that $p(\mathbf{S}, \mathbf{L}^*)$ is $L$-smooth with respect to $\mathbf{S}$, and $L = 4\nu^2$.

Since $p(\mathbf{S}, \mathbf{L})$ is symmetric in $\mathbf{S}$ and $\mathbf{L}$, by similar proof for $\mathbf{L}$, we can show that $p(\mathbf{S}^*, \mathbf{L})$ is $\mu$ strongly-convex and $L$-smooth with respect to $\mathbf{L}$ too. $\square$

### D.2 Proof of Lemma C.2

In this subsection, we prove the first-order stability lemmas.

*Proof.* Take derivative of $p(\mathbf{S}, \mathbf{L})$ with respect to $\mathbf{S}$ while fixing $\mathbf{L}$, we have

$$\nabla_1 p(\mathbf{S}, \mathbf{L}) = \mathbf{\Sigma}^* - (\mathbf{S} + \mathbf{L})^{-1}.$$

Therefore, we have

$$\|\nabla_1 p(\mathbf{S}, \mathbf{L}) - \nabla_1 p(\mathbf{S}, \mathbf{L}^*)\|_F \leq \|(\mathbf{S} + \mathbf{L})^{-1} - (\mathbf{S} + \mathbf{L}^*)^{-1}\|_F. \tag{D.6}$$

We define $\mathbf{\Theta}^* = \mathbf{S} + \mathbf{L}^*, \mathbf{\Theta} = \mathbf{S} + \mathbf{L}$ and $\mathbf{\Delta} = \mathbf{\Theta}^* - \mathbf{\Theta} = \mathbf{L}^* - \mathbf{L}$. Then we have

$$\|(\mathbf{S} + \mathbf{L})^{-1} - (\mathbf{S} + \mathbf{L}^*)^{-1}\|_F = \|(\mathbf{\Theta}^* - \mathbf{\Delta})^{-1} - \mathbf{\Theta}^{*-1}\|_F.$$

Since $\|\boldsymbol{\Theta}^{*-1}\boldsymbol{\Delta}\|_F \leq 1$, we have the convergent matrix expansion

$$(\boldsymbol{\Theta}^* - \boldsymbol{\Delta})^{-1} = \left[\boldsymbol{\Theta}^*(\mathbf{I} - \boldsymbol{\Theta}^{*-1}\boldsymbol{\Delta})\right]^{-1} = \sum_{k=0}^{\infty}(\boldsymbol{\Theta}^{*-1}\boldsymbol{\Delta})^k \boldsymbol{\Theta}^{*-1}.$$

Define $\mathbf{J} = \sum_{k=0}^{\infty}\left(\boldsymbol{\Theta}^{*-1}\boldsymbol{\Delta}\right)^k$, we have

$$\|(\boldsymbol{\Theta}^* - \boldsymbol{\Delta})^{-1} - \boldsymbol{\Theta}^{*-1}\|_F = \left\|\sum_{k=1}^{\infty}(\boldsymbol{\Theta}^{*-1}\boldsymbol{\Delta})^k \boldsymbol{\Theta}^{*-1}\right\|_F = \left\|(\boldsymbol{\Theta}^{*-1}\boldsymbol{\Delta})\mathbf{J}\boldsymbol{\Theta}^{*-1}\right\|_F \leq \|\boldsymbol{\Theta}^{*-1}\|_2^2 \cdot \|\boldsymbol{\Delta}\|_F \cdot \|\mathbf{J}\|_2,$$

(D.7)

where we use the properties of matrix norm that $\|\mathbf{AB}\|_F \leq \|\mathbf{A}\|_2 \cdot \|\mathbf{B}\|_F$ and $\|\mathbf{AB}\|_2 \leq \|\mathbf{A}\|_2 \cdot \|\mathbf{B}\|_2$. By sub-multiplicativity of matrix norm, we have

$$\|\mathbf{J}\|_2 \leq \sum_{k=0}^{\infty}\|\boldsymbol{\Theta}^{*-1}\boldsymbol{\Delta}\|_2^k \leq \frac{1}{1 - \|\boldsymbol{\Theta}^{*-1}\boldsymbol{\Delta}\|_2} \leq \frac{1}{1 - \|\boldsymbol{\Theta}^{*-1}\|_2\|\boldsymbol{\Delta}\|_2} \leq 2. \qquad \text{(D.8)}$$

Note that we have $\|\boldsymbol{\Theta}^{*-1}\|_2 = \lambda_{\max}(\boldsymbol{\Theta}^{*-1}) = (\lambda_{\min}(\boldsymbol{\Theta}^*))^{-1}$. For any $\mathbf{x} \in \mathbb{R}^d$, we have

$$\begin{aligned}
\lambda_{\min}(\boldsymbol{\Theta}^*) &= \min_{\|\mathbf{x}\|_2=1} \mathbf{x}^\top(\mathbf{S} + \mathbf{L}^*)\mathbf{x} \\
&\geq \min_{\|\mathbf{x}\|_2=1} \left\{ -|\mathbf{x}^\top(\mathbf{S} - \mathbf{S}^*)\mathbf{x}| + \mathbf{x}^\top(\mathbf{S}^* + \mathbf{L}^*)\mathbf{x} \right\} \\
&\geq -\|\mathbf{S} - \mathbf{S}^*\|_2 + \lambda_{\min}(\boldsymbol{\Omega}^*) \\
&\geq 1/\nu - R > \frac{1}{2\nu},
\end{aligned}$$

(D.9)

where we use the fact that $\|\mathbf{S} - \mathbf{S}^*\|_2 \leq \|\mathbf{S} - \mathbf{S}^*\|_F \leq R$, $\lambda_{\min}(\boldsymbol{\Omega}_1^*) \geq 1/\nu$ by Assumption 4.1 and $R \leq 1/(2\nu)$. Combining (D.7), (D.8) and (D.9), we have

$$\|(\boldsymbol{\Theta} + \boldsymbol{\Delta})^{-1} - \boldsymbol{\Theta}^{-1}\|_F \leq (2\nu)^2 \cdot \boldsymbol{\Delta} \cdot 2 \leq 8\nu^2\|\mathbf{L} - \mathbf{L}^*\|_F, \qquad \text{(D.10)}$$

which ends the proof. The proof for first-order stability of $\nabla_{\mathbf{L}}p(\mathbf{S}, \mathbf{L})$ is similar and omitted here. $\square$

# E  Auxiliary Lemmas

**Lemma E.1.** [28] Let $f$ be $\mu$-strongly convex and $L$-smooth. Then for any $\mathbf{x}, \mathbf{y} \in \mathbf{dom}f$, we have

$$\langle \nabla f(\mathbf{x}) - \nabla f(\mathbf{y}), \mathbf{x} - \mathbf{y} \rangle \geq \frac{\mu L}{L + \mu}\|\mathbf{x} - \mathbf{y}\|_2^2 + \frac{1}{\mu + L}\|\nabla f(\mathbf{x}) - \nabla f(\mathbf{y})\|_2^2.$$

**Lemma E.2.** [29] Suppose that $\boldsymbol{X}_1, \ldots, \boldsymbol{X}_n \in \mathbb{R}^d$ are i.i.d. sub-Gaussian random vectors. Let $\boldsymbol{\Sigma}^* = \mathbb{E}[1/n \sum_{i=1}^n \boldsymbol{X}_i \boldsymbol{X}_i^\top]$, and we have that

$$\left\|\frac{1}{n}\sum_{i=1}^n \boldsymbol{X}_i \boldsymbol{X}_i^\top - \boldsymbol{\Sigma}^*\right\|_{\infty,\infty} \leq 2\max_i \Sigma_{ii}^* \sqrt{\frac{\log d}{n}}$$

holds with probability at least $1 - C/d$, where $C > 0$ is a constant.

**Lemma E.3.** [32] Suppose that $\boldsymbol{X}_1, \ldots, \boldsymbol{X}_n \in \mathbb{R}^d$ are i.i.d. sub-Gaussian random vectors. Let $\boldsymbol{\Sigma}^* = \mathbb{E}[1/n \sum_{i=1}^n \boldsymbol{X}_i \boldsymbol{X}_i^\top]$, and we have that

$$\left\|\frac{1}{n}\sum_{i=1}^n \boldsymbol{X}_i \boldsymbol{X}_i^\top - \boldsymbol{\Sigma}^*\right\|_2 \leq 2\lambda_{\max}(\boldsymbol{\Sigma}^*)\sqrt{\frac{d}{n}}$$

holds with probability at least $1 - C'/d$, where $C' > 0$ is a constant.

**Lemma E.4.** [31] For any $\mathbf{Z}, \mathbf{Z}^* \in \mathbb{R}^{d \times r}$, we have

$$d(\mathbf{Z}, \mathbf{Z}^*) \leq \frac{1}{\sqrt{2(\sqrt{2} - 1)}\sigma_{\min}(\mathbf{Z}^*)}\|\mathbf{Z}\mathbf{Z}^\top - \mathbf{Z}^*\mathbf{Z}^{*\top}\|_F,$$

where $\sigma_{\min}(\mathbf{Z}^*)$ is the minimal nonzero singular value of $\mathbf{Z}^*$.

# F    Additional Experiments on Cancer Genomic Data

To further show the performances of different methods on recovering the edges in the benchmark network that are most related to luminal breast cancer, we chose the top 50 gene pairs with highest regulatory potential scores based on the Cistrome Cancer Database, and plotted the edges identified by each method in Figure 3. Note that the estimated networks of methods based on LVGGM (**ADMM**, **PPA** and **AltGD**) have much more overlaps with the benchmark network on the top 50 edges than **GLasso**, which ignores the latent structure of precision matrix.

Figure 3: A comparison between the inferred regulatory network as compared to the regulatory potential score from the Cistrome Cancer Database on luminal breast cancer. We chose the top 50 gene pairs in the Database with highest regulatory potential scores.

We also plotted the regulatory potential scores for basal subtype breast cancer based on Cistrome Cancer Database in Figure 4. We can see that the estimated networks of **ADMM**, **PPA** and **AltGD** again have much more overlaps with the benchmark network on the top 50 edges than **GLasso**, which is consistent with the results for luminal breast cancer.

Figure 4: A comparison between the inferred regulatory network as compared to the regulatory potential score from Cistrome Cancer Database on basal breast cancer. We chose the top 50 gene pairs in Cistrome Cancer Database with highest regulatory potential scores.