[Reviews · NeurIPS 2017]

Reviewer 1



The authors present an efficient algorithm for latent variable Gaussian graphical models (LVGGMs) in which the precision matrix of the observations is the sum of a sparse and a low rank matrix. Their approach is based on matrix factorizations while previous methods were based on nuclear norm penalized convex optimization schemes. The advantage of this approach is that the resulting nonconvex optimization problem is much easier to optimize. The authors prove that their algorithm (basically an alternating gradient method for the sparse and low rank pieces separately) has a linear rate of convergence in a region of the global optimum. The authors provide an initialization method that produces a good initialization with high probability, and the authors conclude with an experimental evaluation that compares their method to the competing convex optimization methods (their algorithm often performs comparably or better while running much faster than the competing methods). The paper is well-written but feels defensive in parts. - Just a thought: I recall reading somewhere that estimating the sample precision matrix by inverting the sample covariance matrix is a bad thing to do, especially in higher dimensions. It may be worthwhile to replace the inverse with something else (perhaps something that doesn't have the same O(d^3) complexity). - A variety of fast SVD methods that approximately compute the full SVD are available. It would be more interesting to compare to the exact methods where these additive approximations have been swapped in. I believe that they often have much less than O(d^3) complexity while still providing a quality approximation.

Reviewer 2



The paper considers learning the dependency structure of Gaussian graphical models where some variables are latent. Directly applying the usual assumption of sparsity in the precision matrix is difficult because variables that appear correlated might actually both depend on a common latent variable. Previously, Chandrasekaran et al. proposed estimating the model structure by decomposing the full precision matrix into the sum of of a sparse matrix and a low-rank matrix. Likelihood is maximized while the components of the sparse matrix are penalized with an l1 regularizer and the low-rank matrix is penalized with a nuclear norm. Computing the proximal operator to update the low-rank component requires performing SVD in O(d^3) time at each iteration. The authors propose replacing the low-rank component with its Cholesky decomposition ZZ^T and finding Z directly. The main technical contribution is that they prove linear converge of an alternating gradient algorithm, up to optimal statistical precision. Experiments show that by avoiding the need for SVD at each iteration, the proposed method is orders of magnitude faster than the method of Chandrasekaran et al. or a related approach that uses ADMM to estimate the model. It is also more accurate on synthetic and gene network data. Is Huang, Jianhua Z., et al. "Covariance matrix selection and estimation via penalised normal likelihood." Biometrika 93.1 (2006): 85-98 related? They propose regularizing the Cholesky factorization for precision matrix estimation, although without any latent variables. Their motivation is that it avoids needing to check that the estimated precision matrix is PSD, which takes O(d^3) time.

Reviewer 3



The contribution is interesting but not on the top of the NIPS list. The paper builds a non-convex estimator for a latent variable Gaussian Graphical Model under structural assumption that the resulting precision matrix is a superposition of a sparse matrix and a low rank matrix, and then suggest an empirical iterative algorithm. Based on numerical computations the authors claim the algorithm is fast and efficient. However, it is not clear to the reviewer why only experiments with a relatively small latency with, $r$, much smaller than $d$ is considered. The algorithm would certainly be much more sounds if extensions to the case with $r\sim d$ would be claimed. Wrt the theory. The main theory statement on the algorithm convergence is stated under a number of rather non-intuitive assumptions, which dilute the results. It would be useful to get some intuition and clarification on the nature of the imposed conditions.